
# Limiting amplitudes of fully nonlinear interfacial tides and solitons

Borja Aguiar-González[1,*] and Theo Gerkema[2]

[1]Departamento de Física, Facultad de Ciencias del Mar, Universidad de Las Palmas de Gran Canaria, E-35017 Las Palmas, Spain.
[2]NIOZ Royal Netherlands Institute for Sea Research, P.O. Box 59, 1790 AB Den Burg, Texel, The Netherlands
[*]now at: NIOZ Royal Netherlands Institute for Sea Research, P.O. Box 59, 1790 AB Den Burg, Texel, The Netherlands

*Correspondence to:* B. Aguiar-González
(aguiar@nioz.nl)

**Abstract.** A new two-fluid layer model consisting of forced rotation-modified Boussinesq equations is derived for studying tidally-generated fully nonlinear, weakly nonhydrostatic dispersive interfacial waves. This set is a generalization of the Choi-Camassa equations, extended here with forcing terms and Coriolis effects. The forcing is represented by a horizontally oscillating sill, mimicking

5     a barotropic tidal flow over topography. Solitons are generated by a disintegration of the interfacial tide. Because of strong non-linearity, solitons in some cases attain a limiting table-shaped form, in accordance with soliton theory. More generally, we use the model equations to investigate the role of the initial stages of the internal tide on the limiting amplitudes of solitons under fully nonlinear conditions. Numerical solutions reveal that the tide-generated solitons are primarily limited by the

10     underlying quasi-nonlinear internal tide. We show the decisive factor is the generation of higher harmonics, which already limit the growth of the initial internal tide. As a consequence, and contrary to predictions by classical KdV theory alone, we find that tidally generated solitons are subjected to limiting amplitudes even under weakly nonlinear conditions. This implies that under strongly nonlinear conditions, amplitudes of solitons may be limited before attaining a table-shaped form.

15     ## 1 Introduction

Tidally-generated internal solitons are a widespread phenomenon in the oceans and they have been observed and studied for decades (see, e.g., Apel et al. (2006)). They are intrinsically linked to the internal tide, which itself is generated by barotropic tidal flow over topography. As the internal tide steepens, it may split up into groups of internal solitons, which therefore appear at the tidal period.



For internal solitons as such, an archetypal model has been the Korteweg-de Vries (KdV) equation, which is based on the assumption of weak nonlinearity and weak nonhydrostatic effects. The equation gives prediction for the relation between amplitude, width and phase speed of the solitons, as well as the shape itself. In the KdV equation there is, mathematically speaking, no limit to the amplitude that solitons may reach (although, of course, at some point the underlying assumption of weak nonlinearity would be violated). This behaviour changes fundamentally if a higher-order (i.e., cubic) nonlinear term is included, leading to the so-called extended KdV (eKdV) equation, as discussed in, e.g., Helfrich and Melville (2006). This extended version produces qualitatively different solitons: their amplitude is limited (for a given configuration of layers) and they broaden as they reach their maximum amplitude, the so-called table-top solitons. This behaviour is confirmed by the fully nonlinear soliton models, as derived by Choi and Camassa (1999) and Miyata (1985, 1988) (denoted as the MCC equations for brevity).

In this paper, we focus on another limiting factor, which comes into play even before solitons arise, namely in the internal tide itself. In a purely linear system, the amplitude of the internal tide increases linearly with the strength of the barotropic tidal flow. Here we study how this changes if one includes quasi-nonlinear effects (i.e., products of barotropic and baroclinic terms) and genuinely nonlinear effects (products of baroclinic terms). We demonstrate that a saturation in the amplitude of the internal tide occurs in both cases; increasing the barotropic flow further does not produce a larger internal tide. As a consequence, resulting solitons may stay well below their formal limiting amplitude, no matter how strong the forcing.

To study these effects, we derive a set of fully nonlinear, weakly nonhydrostatic model equations, by extending the MCC equations with a barotropic tidal forcing over topography and with Coriolis effects, which have previously been shown to play a key role in soliton generation from internal tides (Gerkema and Zimmerman, 1995). To avoid nonlinearities in the barotropic flow itself (which cannot be formally neglected in a fully nonlinear model), we mimic the generation by barotropic tidal flow over topography with a horizontally oscillating topography. (There is no complete equivalence, but we demonstrate that for the parameters used here, the difference remains small.) The presence of a topography greatly complicates the subsequent handling of the equation, necessary to bring them in a form amenable to numerical solving, but we demonstrate that the set of equations can be obtained.

An extension of the MCC theory with Coriolis effects (MCC-f) was already derived by Helfrich (2007), who investigated on the decay and return of internal solitary waves with rotation. We focus on the novel aspect of studying the wave evolution and limiting amplitudes of fully nonlinear,



weakly nonhydrostatic internal tides and solitons when a forcing, and rotational effects, are acting. We denote our extension of the MCC theory as forced-MCC-f (forced-MCC in absence of rotation), for brevity.


The paper is organized as follows. We derive a new two-fluid layer model consisting of a set of forced rotation-modified Boussinesq equations in Sect. 2. We start with the basic equations and assumptions. Then, we scale equations (Sect. 2.1) and vertically integrated them over the layers (Sect. 2.2). Up to this point, the resulting equations are exact but do not form a closed set. The set

is closed by making an expansion in a small parameter measuring the strength of non-hydrostaticity (Sect. 2.3). The resulting model turns out equivalent to Choi-Camassa equations plus additional terms which provide the forcing and rotation effects to the system. In Sect. 3, we discuss the problem of having an oscillating topography instead of a barotropic flow over topography.

Numerical solutions of the model to study the limiting amplitudes of the internal tides and solitons are presented in Sect. 4. First, the non-rotating quasi-nonlinear and fully-nonlinear cases are explored (Sects. 4.1 and 4.2, respectively). Second, rotational effects related to limiting amplitudes of tidally-generated fully nonlinear solitons are investigated (Sect. 4.3). A summary and main conclusions are presented in Sect. 5. The numerical methods and schemes are described in *Appendix A*.

The actual form of the model equations as used in the code is presented in *Appendix B*.

## 2   Derivation of the model

We start from the continuity and Euler equations and consider a two-fluid layer system (Fig. 1) with a jump in density accross the interface and in which each layer is composed of a homogeneous,

inviscid, and incompressible fluid, applying the Boussinesq approximation. We also assume uniformity in one of the horizontal directions, taking $\partial/\partial y = 0$. Hence, the continuity and momentum equations read

$$u_{i,x} + w_{i,z} = 0 \tag{1}$$

$$\bar{\rho}\Big(u_{i,t} + u_i\, u_{i,x} + w_i\, u_{i,z} - f\, v_i\Big) = -p_{i,x} \tag{2}$$

$$v_{i,t} + u_i\, v_{i,x} + w_i\, v_{i,z} + f\, u_i = 0 \tag{3}$$

$$\bar{\rho}\Big(w_{i,t} + u_i\, w_{i,x} + w_i\, w_{i,z}\Big) = -p_{i,z} - \rho_i\, g \tag{4}$$

where $\rho_i$ is density, $(u_i, v_i, w_i)$ are the velocity components in Cartesian coordinates, $p_i$ is pressure, $g$ the gravitational accelaration, $f$ the Coriolis parameter ($f = 2\Omega\sin\phi$, at latitude $\phi$) and $\bar{\rho}$ the mean density. The subscript $i = 1$ ($i = 2$) refers to the upper (lower) layer and a stable stratification,

$\rho_1 < \rho_2$, is assumed.





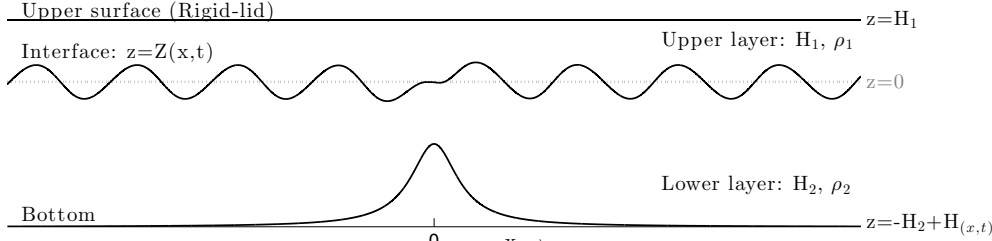

**Fig. 1.** The two-fluid layer system for which the forced-MCC-f equations are derived. The horizontal dashed grey line indicates the level $z = 0$, the level at which the interface resides at rest.

Boundaries are defined at the surface, taken to be a rigid lid, which is located at $z = H_1$, and at the bottom, located at $z = -H_2 + H(x, t)$. The time-dependence of the bottom will later be specified as a horizontal oscillation, mimicking a barotropic tidal flow over topography, the forcing mechanism for internal tides. However, the two are not exactly equivalent, since the transformation from one frame of reference to the other involves an acceleration and is therefore not Galilean. We further discuss this aspect in Sect. 3.

The kinematic boundary conditions at the surface and bottom read

$$w_1 = 0 \qquad \text{at } z = H_1 \tag{5}$$

$$w_2 = H_t + H_x\, u_2 \qquad \text{at } z = -H_2 + H(x, t). \tag{6}$$

At the interface, $z = Z(x, t)$, which if at rest is located at $z = 0$, the boundary conditions are given by the continuity of normal velocity and pressure:

$$w_i = Z_t + u_i\, Z_x \text{ and } p_1 = p_2 \quad \text{at } z = Z. \tag{7}$$

For later convenience, we write pressure as the sum of hydrostatic and dynamic parts, the latter being denoted by primes:

$$p_i = \rho_1 g H_1 - \rho_i g z + p_i'(t, x, z).$$

In the horizontal momentum equation, this amounts to replacing $p_{i,x}$ by $p_{i,x}'$, whereas the vertical momentum equation (4) gives

$$\bar{\rho}\Big(w_{i,t} + u_i\, w_{i,x} + w_i\, w_{i,z}\Big) = -p_{i,z}'.$$

Continuity of pressure at the interface, the second equation in (7), now becomes

$$(p_1' - p_2')|_{z=Z} = (\rho_1 - \rho_2)g Z.$$



### 2.1 Scaling

The next step is to bring the equations into an appropriate dimensionless form for which we intro-
duce the following scales. The scale for the undisturbed water depth is taken to be $D$, and the typical
wavelength $L$. Crucially, we will assume waves to be long, i.e. nonhydrostatic effects to be weak.
This will be expressed by the small parameter[1], $\delta = \left(\frac{D}{L}\right)^2 \ll 1$.

Since we allow waves to have large amplitudes (i.e. being strongly nonlinear), we may take horizon-
tal current velocities to scale with $c_0 = (g'D)^{1/2}$, where $g'$ is reduced gravity, $g' = g\,(\rho_2 - \rho_1)/\bar{\rho}$;
and, $c_0$ is an approximate measure of the linear long-wave phase speed for interfacial waves. Thus,
$u$ and $v$ will be scaled with $c_0$. For the interfacial displacement being allowed to be large, an appro-
priate scale of $Z$ is $D$.

The typical scale of $w$ now follows from the continuity equation as $Dc_0/L$. Finally, the scale of
pressure follows from assuming a primary balance between the acceleration term $\bar{\rho}\,u_t$ and $p_x$ in the
horizontal momentum equation.

In summary, then, we can introduce the following dimensionless variables, indicated by asterisks,

$$x = L\,x^*, \quad z = D\,z^*, \quad t = (L/c_0)\,t^*,$$
$$p'_i = (\bar{\rho}\,c_0^2)\,p'^*_i, \quad u_i = c_0\,u_i^*, \quad v_i = c_0\,v_i^*, \quad w_i = (D/L)\,c_0\,w_i^*. \tag{8}$$

With these scales, the dimensionless continuity and Euler equations yield (for convenience, we drop
the asterisks right away):

$$u_{i,x} + w_{i,z} = 0 \tag{9}$$

$$u_{i,t} + u_i\,u_{i,x} + w_i\,u_{i,z} - \mu\,v_i = -p'_{i,x} \tag{10}$$

$$v_{i,t} + u_i\,v_{i,x} + w_i\,v_{i,z} + \mu\,u_i = 0 \tag{11}$$

$$\delta\Big(w_{i,t} + u_i\,w_{i,x} + w_i\,w_{i,z}\Big) = -p'_{i,z}. \tag{12}$$

Here $\mu$ is the scaled Coriolis parameter, $\mu = fL/c_0$. Furthermore we introduce the dimensionless
quantities $\zeta$, $h_i$, and $h$ via $(Z, H_1, H_2, H) = D(\zeta, h_1, h_2, h)$, so that the scaled form of the boundary
conditions is

$$w_1 = 0 \quad \text{at } z = h_1 \tag{13}$$

$$w_i = \zeta_t + u_i\,\zeta_x \quad \text{at } z = \zeta(x,t) \tag{14}$$

$$p'_2 - p'_1 = \zeta \quad \text{at } z = \zeta(x,t) \tag{15}$$

$$w_2 = h_t + u_2\,h_x \quad \text{at } z = -h_2 + h(x,t)\,. \tag{16}$$

---

[1]In Choi and Camassa (1999) a small parameter $\epsilon$ was used instead, which relates to ours as $\delta = \epsilon^2$.





The goal is now to derive a reduced set of equations from (9)–(12), in which the boundary conditions
(13)–(16) are incorporated by vertical integration, exploiting the smallness of the parameter $\delta$. The
procedure is identical to that of (Choi and Camassa, 1999), but with the additional complications of
the Coriolis force, topography, and tidal forcing.

### 2.2 Vertically integrated equations

We vertically integrate the equations over the upper and lower layers and expand them to the or-
ders $\delta^0$ and $\delta^1$ to obtain a closed set for the weakly nonhydrostatic equations, following Choi and
Camassa (1999). The layer-mean $\bar{f}_1$ of a function $f_1(x,z,t)$ for the upper layer is being defined as

$$\bar{f}_1(x,t) = \frac{1}{\eta_1} \int_\zeta^{h_1} dz\, f_1(x,z,t), \quad \eta_1 = h_1 - \zeta \tag{17}$$

and for the lower layer as

$$\bar{f}_2(x,t) = \frac{1}{\eta_2} \int_{-h_2+h}^{\zeta} dz\, f_2(x,z,t), \quad \eta_2 = h_2 - h + \zeta. \tag{18}$$

where $\eta_i$ represents the thickness of the layer (depending on the interfacial displacement $\zeta$). Notice
that the boundaries of the integral depend on time and space ($x$) via the interfacial movement $\zeta(t,x)$,
but also, for the lower layer, via the horizontally oscillating topography $h(t,x)$.[2] Before proceeding,
nonlinear terms in horizontal momentum equations (10) and (11) are rewritten as $(u_i^2)_x + (w_i u_i)_z$
and $(u_i v_i)_x + (w_i v_i)_z$, respectively, to facilitate the procedure.

After integration of Eqs. (9)–(11) for $i = 1$ and applying the boundary conditions (13)–(15) we
obtain the layer-mean equations for the upper layer

$$\eta_{1,t} + (\eta_1 \bar{u}_1)_x = 0, \tag{19}$$

$$(\eta_1 \bar{u}_1)_t + (\eta_1 \overline{u_1 u_1})_x - \mu \eta_1 \bar{v}_1 = -\eta_1 \overline{p'_{1,x}}, \tag{20}$$

$$(\eta_1 \bar{v}_1)_t + (\eta_1 \overline{u_1 v_1})_x + \mu \eta_1 \bar{u}_1 = 0. \tag{21}$$

For the lower layer one proceeds similarly, except that now both boundaries are variable. Applying
the boundary conditions (14)–(16), vertical integration of (9)–(11) for $i = 2$ yields

$$\eta_{2,t} + (\eta_2 \bar{u}_2)_x = 0, \tag{22}$$

$$(\eta_2 \bar{u}_2)_t + (\eta_2 \overline{u_2 u_2})_x - \mu \eta_2 \bar{v}_2 = -\eta_2 \overline{p'_{2,x}}, \tag{23}$$

$$(\eta_2 \bar{v}_2)_t + (\eta_2 \overline{u_2 v_2})_x + \mu \eta_2 \bar{u}_2 = 0. \tag{24}$$

---

[2]For this reason we need to apply the Leibniz integral rule below with respect to $x$ and $t$.



### 2.3 Expansion in $\delta$

The six integrated equations (19)–(24) derived so far are exact but do not form a closed set. The variables $\eta_1$, $\eta_2$ and $\zeta$ count as one unknown, but we have also $\bar{u}_i$, $\bar{v}_i$, $\overline{p'_{i,x}}$, $\overline{u_i u_i}$ and $\overline{u_i v_i}$, giving 11 unknowns for 6 equations. To obtain a closed set, the last two expressions will be cast in terms of $\bar{u}_i$ and $\bar{v}_i$ by using the vertical momentum equation, expanded in terms of the small parameter $\delta$.

Furthermore, continuity of pressure at the interface is used to connect the pressure in the lower and upper layer (i.e., $\overline{p'_{1,x}}$ and $\overline{p'_{2,x}}$). All in all, the six equations are thus modified to contain only six unknowns. With this aim, we make a formal expansion of the unknowns for the lowest ($\delta^0$) and next ($\delta$) orders, as, for example:

$$\bar{f}_i = \bar{f}_i^{(0)} + \delta \bar{f}_i^{(1)} + \cdots$$

At the lowest order ($\delta^0$), $p'^{(0)}$ accounts for hydrostatic effects. At the next order ($\delta$), $p'^{(1)}$ brings weakly nonhydrostatic effects into the system.

#### 2.3.1 Lowest order

At lowest order, the vertical momentum equation (12) reduces to $\partial p_i'^{(0)}/\partial z = 0$ as terms of order $\delta$

are neglected; therefore, (perturbation) pressure is vertically constant in each layer. For convenience, we introduce $P = p_2'^{(0)}$, being a function of $t$ and $x$. It then follows from continuity of pressure at the interface, that $p_1'^{(0)} = P - \zeta$. Thus, to this order of approximation,

$$\overline{p'_{1,x}} = P_x - \zeta_x + O(\delta), \tag{25}$$

and for the lower layer

$$\overline{p'_{2,x}} = P_x + O(\delta). \tag{26}$$

Returning to the original horizontal momentum equations, it is now natural to assume that the horizontal velocities, too, are independent of $z$ within each layer; then, given the $z$-independence of pressure

$$\overline{u_i u_i} = \bar{u}_i^2 + O(\delta), \qquad \overline{u_i v_i} = \bar{u}_i \bar{v}_i + O(\delta).$$

At lowest order, then, the set of integrated equations is closed; together with the (exact) integrated continuity equations (19) and (22), we have the momentum equations in terms of the six variables $\bar{u}_i$, $\bar{v}_i$, $\zeta$ and $P$:

$$(\eta_1 \bar{u}_1)_t + (\eta_1 \bar{u}_1^2)_x - \mu \eta_1 \bar{v}_1 = -\eta_1 (P_x - \zeta_x) + O(\delta), \tag{27}$$

$$(\eta_2 \bar{u}_2)_t + (\eta_2 \bar{u}_2^2)_x - \mu \eta_2 \bar{v}_2 = -\eta_2 P_x + O(\delta), \tag{28}$$

$$(\eta_1 \bar{v}_1)_t + (\eta_1 \bar{u}_1 \bar{v}_1)_x + \mu \eta_1 \bar{u}_1 = O(\delta), \tag{29}$$

$$(\eta_2 \bar{v}_2)_t + (\eta_2 \bar{u}_2 \bar{v}_2)_x + \mu \eta_2 \bar{u}_2 = O(\delta). \tag{30}$$




Recall that $\eta_{1,2}$ can be expressed in terms of $\zeta$ and thus involve just one unknown.

### 2.3.2 Next order

To include terms of order $\delta$, the key problem is, again, to close the set of six vertically integrated equations by deriving closed expressions for the horizontal pressure gradients $\overline{p'_{i,x}}$ as well as for the contributions of $\overline{u_i u_i}$ and $\overline{u_i v_i}$ in the nonlinear terms. The latter problem is particularly simple. At order $\delta$, the products contain one lowest-order field, which is independent of $z$ (e.g., $u_i^{(0)} = \bar{u}_i^{(0)}$), hence

$$\overline{u_i u_i} = \frac{1}{\eta_i} \int dz\, u_i^2 = \frac{1}{\eta_i} \int dz\, (u_i^{(0)\,2} + 2\delta u_i^{(0)} u_i^{(1)} + \cdots)$$

$$= \bar{u}_i^{(0)\,2} + 2\delta \bar{u}_i^{(0)} \bar{u}_i^{(1)} + \cdots$$

$$= \bar{u}_i^2 + O(\delta^2)$$

so that

$$\overline{u_i u_i} = \bar{u}_i^2 + O(\delta^2), \qquad \overline{u_i v_i} = \bar{u}_i \bar{v}_i + O(\delta^2).$$

This allows us to write the horizontal momentum equations as

$$(\eta_1 \bar{u}_1)_t + (\eta_1 \bar{u}_1^2)_x - \mu \eta_1 \bar{v}_1 = -\eta_1 \overline{(p'_1{}^{(0)} + \delta p'_1{}^{(1)})_x} + O(\delta^2) \tag{31}$$

$$(\eta_2 \bar{u}_2)_t + (\eta_2 \bar{u}_2^2)_x - \mu \eta_2 \bar{v}_2 = -\eta_2 \overline{(p'_2{}^{(0)} + \delta p'_2{}^{(1)})_x} + O(\delta^2) \tag{32}$$

$$(\eta_1 \bar{v}_1)_t + (\eta_1 \bar{u}_1 \bar{v}_1)_x + \mu \eta_1 \bar{u}_1 = O(\delta^2) \tag{33}$$

$$(\eta_2 \bar{v}_2)_t + (\eta_2 \bar{u}_2 \bar{v}_2)_x + \mu \eta_2 \bar{u}_2 = O(\delta^2) \tag{34}$$

The remaining problem is to find an expression for $p'_i{}^{(1)}$. At order $\delta$, Eq. (12) reads in terms of the lowest order vertical velocities,

$$w_i^{(0)}{}_t + u_i^{(0)} w_i^{(0)}{}_x + w_i^{(0)} w_i^{(0)}{}_z = -p'_i{}^{(1)}{}_z \tag{35}$$

From vertically integrating the continuity equation (9), we obtain an expression for $w_i^{(0)}$:

$$w_i^{(0)} = -z \bar{u}_{i,x}^{(0)} + c_i(t,x)$$

where $c_i$ are 'constants' of integration which are determined by using the boundary conditions at the surface (13) and bottom (16). Thus, $w_i^{(0)}$ for the upper- and lower layers become, respectively,

$$w_1^{(0)} = (h_1 - z)\, \bar{u}_{1,x}^{(0)}, \tag{36}$$

$$w_2^{(0)} = (h - h_2 - z)\, \bar{u}_{2,x}^{(0)} + D_2 h, \tag{37}$$

where the operator $D_i$ is defined as $\partial/\partial t + \bar{u}_i^{(0)} \partial/\partial x$. Substituting $w_1^{(0)}$ from Eq. (36) and $w_2^{(0)}$
from Eq. (37) into Eq. (35), and vertically integrating the result, we get an expression for $p'_1{}^{(1)}$ and



$p_2'^{(1)}$. Taking their derivative with respect to $x$ and their mean over each layer, we finally obtain an expression for $\overline{p_{i,x}'^{(1)}}$ at the upper and lower layer at order $\delta$. Including the lowest order terms (25) and (26), this allow us to write the horizontal pressure gradient for the upper layer

$$\overline{p_{1,x}'} = \overline{p_{1,x}'^{(0)}} + \delta \overline{p_{1,x}'^{(1)}} + O(\delta^2) = P_x - \zeta_x - \delta \Big[ \frac{1}{3\eta_1}(\eta_1^3 G_1)_x \Big] + O(\delta^2) \,, \tag{38}$$

and, for the lower layer,

$$\overline{p_{2,x}'} = \overline{p_{2,x}'^{(0)}} + \delta \overline{p_{2,x}'^{(1)}} + O(\delta^2) = P_x - \delta \Big[ \frac{1}{3\eta_2}(\eta_2^3 G_2)_x + \frac{1}{2}\eta_2 G_2 h_x - \frac{\eta_2}{2}(D_2^2 h)_x - \zeta_x D_2^2 h \Big] + O(\delta^2), \tag{39}$$

where we introduced for simplicity the term $G_i$ (as in Choi and Camassa (1999)),

$$G_i = \bar{u}_{i,xt}^{(0)} + \bar{u}_i^{(0)} \bar{u}_{i,xx}^{(0)} - (\bar{u}_{i,x}^{(0)})^2 \,. \tag{40}$$

With this, the horizontal momentum equations (31) and (32) become

$$(\eta_1 \bar{u}_1)_t + (\eta_1 \bar{u}_1^2)_x - \mu \eta_1 \bar{v}_1 = -\eta_1 \Big\{ P_x - \zeta_x - \delta \Big[ \frac{1}{3\eta_1}(\eta_1^3 G_1)_x \Big] \Big\} + O(\delta^2) \tag{41}$$

$$(\eta_2 \bar{u}_2)_t + (\eta_2 \bar{u}_2^2)_x - \mu \eta_2 \bar{v}_2 = -\eta_2 \Big\{ P_x - \delta \Big[ \frac{1}{3\eta_2}(\eta_2^3 G_2)_x + \frac{1}{2}\eta_2 G_2 h_x - \frac{\eta_2}{2}(D_2^2 h)_x - \zeta_x D_2^2 h \Big\} + O(\delta^2) \tag{42}$$

We have thus obtained a closed set of six dimensionless equations, namely the exact continuity equa-
tions (19) and (22), the horizontal momentum equations (41) and (42), as well as (33) and (34); the last four equations involve the the weakly non-hydrostatic assumption. The six unknowns are $\bar{u}_1$, $\bar{u}_2$, $\bar{v}_1$, $\bar{v}_2$, $P$, and (via $\eta_{1,2}$) $\zeta$. In the absence of an interfacial wave forcing and neglecting Earth's rotation effects, our set of equations reduces to those of Choi and Camassa (1999).

Before proceeding to numerical solving the set, we further specify the model by prescribing the oscillating topography, i.e., the forcing to the system, with

$$h = h(X) \quad \text{with} \quad X(x,t) = x - a\cos t \quad (a \text{ being an arbitrary positive constant}) \,. \tag{43}$$

We combine the continuity equations (19) and (22) into

$$(\eta_1 + \eta_2)_t + (\eta_1 \bar{u}_1 + \eta_2 \bar{u}_2)_x = 0 \,, \tag{44}$$

Given that $\eta_1 + \eta_2 = h_1 + h_2 - h$, with the two-fluid system depth $h_1 + h_2 = 1$, this leads to

$$-h_t + (\eta_1 \bar{u}_1 + \eta_2 \bar{u}_2)_x = 0. \tag{45}$$

If we now substitute the time derivative of the oscillating topography (43) above, it yields

$$(\eta_1 \bar{u}_1 + \eta_2 \bar{u}_2)_x = U \frac{\partial h}{\partial x} \,, \tag{46}$$





with

$$U = a \sin t, \tag{47}$$

which mimicks a barotropic tidal flow over the oscillating topography (i. e. the velocity of the moving topography). Then, Eq. (46) can be integrated in $x$,

$$\eta_1 \bar{u}_1 + \eta_2 \bar{u}_2 = Uh + A(t), \tag{48}$$

where we assume that initially $\bar{u}_1 = \bar{u}_2 = U = 0$, so that $A(t) = 0$. Notice that the right-hand side is prescribed via the forcing and is thus a known quantity. It allows us to express $\bar{u}_2$ in terms of $\bar{u}_1$. We can thus combine the horizontal momentum equations (41) and (42), eliminating $P$,

$$\bar{u}_{1,t} + \bar{u}_1 \bar{u}_{1,x} + \mu \bar{v}_1 = \zeta_x + \frac{1}{(1-h)}\Big((Uh)_t + (\eta_1 \bar{u}_1^2 + \eta_2 \bar{u}_2^2)_x - \mu(\eta_1 \bar{v}_1 + \eta_2 \bar{v}_2) - \eta_1 \zeta_x\Big) +$$

$$\delta\Big(1 - \frac{\eta_1}{(1-h)}\Big)\Big[\eta_1 G_1 \eta_{1,x} + \frac{\eta_1^2}{3} G_{1,x}\Big]$$


$$+ \frac{\delta \eta_2}{(1-h)}\Big[ -\eta_2 G_2 \zeta_x - \frac{\eta_2^2}{3} G_{2,x} + \frac{\eta_2 G_2}{2} h_x + \frac{\eta_2}{2}(D_2^2 h)_x + \zeta_x D_2^2 h \Big] + O(\delta^2) \tag{49}$$

$$\bar{u}_2 = \frac{Uh - \eta_1 \bar{u}_1}{\eta_2}, \tag{50}$$

$$\bar{v}_{1,t} + \bar{u}_1 \bar{v}_{1,x} + \mu \bar{u}_1 = 0 + O(\delta^2), \tag{51}$$

$$\bar{v}_{2,t} + \bar{u}_2 \bar{v}_{2,x} + \mu \bar{u}_2 = 0 + O(\delta^2), \tag{52}$$


$$\zeta_t - (h_1 - \zeta)\bar{u}_{1,x} + \bar{u}_1 \zeta_x = 0. \tag{53}$$

where the $\bar{v}_i$–horizontal momentum equations (51) and (52) have been further simplified from (33) and (34) by using the continuity equations (19) and (22). Eq. (19) has now been expressed in terms of $\zeta$ for convenience. The other continuity equation (22) is no longer included explicitly since it is already present via (50). All in all, we have now five equations for five unknowns ($\bar{u}_1, \bar{u}_2, \bar{v}_1, \bar{v}_2$ and

$\zeta$).

The model equations are here in nondimensinal form, but for clarity in the next sections we present results from numerical experiments after re-dimensionalisation. The numerical methods and schemes are described in *Appendix* A. The actual form of the model equations as used in the code is presented

in *Appendix B*.

## 3  Preliminary numerical experiments

We discuss here the 'non-inertial' nature of our frame of reference to test the applicability to ocean cases, where the topography is at rest.



A Galilean transformation involves two frames of reference which move with constant and rectilinear speed with respect to each another. Hence, observations made in one frame can be converted to another, as physical laws are identical.

However, our oscillating topography is not an inertial frame since it is accelerated with respect to a

situation where the topography is at rest (as in the ocean). It is, therefore, not evident that the results from the two frames are equivalent.

The topography is prescribed analytically, in order to ensure perfectly smooth second and third derivatives of $h(x,t)$:

$$H(x,t) = \frac{H_T}{1 + (x/H_L)^2} \tag{54}$$

with $x$ being the grid positions in space; and, $H_T$ and $H_L$ being the parameters which set the height and width of a symmetric sill, respectively. Other analytical functions may be also used depending on the desired topography. The topographic obstacle (ridge, sill, ...) is always centred in the $x$-axis and the length of the $x$-domain is chosen to be large enough to prevent that waves from reaching the

boundaries. At this point it is worth while to recall that the oscillation of the topography is included within the model, in dimensionless form, as $h = h(X)$, with $X(x,t) = x - a\cos t$ with $a$ being an arbitrary constant.

In all experiments, fluid starts moving to the right at $t = 0$ (i.e., topography moving to the left). The

waves are generated near the origin in x-axis due to the 'tide-topography' interaction; on the negative (positive) $x$-axis, waves travel to the left (right). Because the forcing enters in the simulation asymmetrically with fluid at rest moving to the right, it is expected that wave packets in the front appear rather different when comparing both sides (negative vs. positive $x$-domain). These fronts are the transients, which are influenced by the way the experiment is started. A steady solution at

both sides of the $x$-axis is reached after several tidal periods have passed away. For this reason, we start the observation of all our results when the signal has become periodic to avoid transient effects.

We use the generation model of weakly nonlinear, weakly nonhydrostatic solitons derived in Gerkema (1996), which works with tidal motion over fixed topography, as a benchmark for testing our model.

Of course, departures may arise from the fact the model derived here is fully nonlinear. However, the effects of nonlinearity come into play more as the waves propagate away, whereas the differences between the frames of reference are expected to be largest exactly over the topography. If the results between the models turn out to be similar over the topography, it thus seems reasonable to conclude that we can compare our present setting to that in the ocean. For this reason, we restrict ourselves to

the linear and quasi-nonlinear regimes.





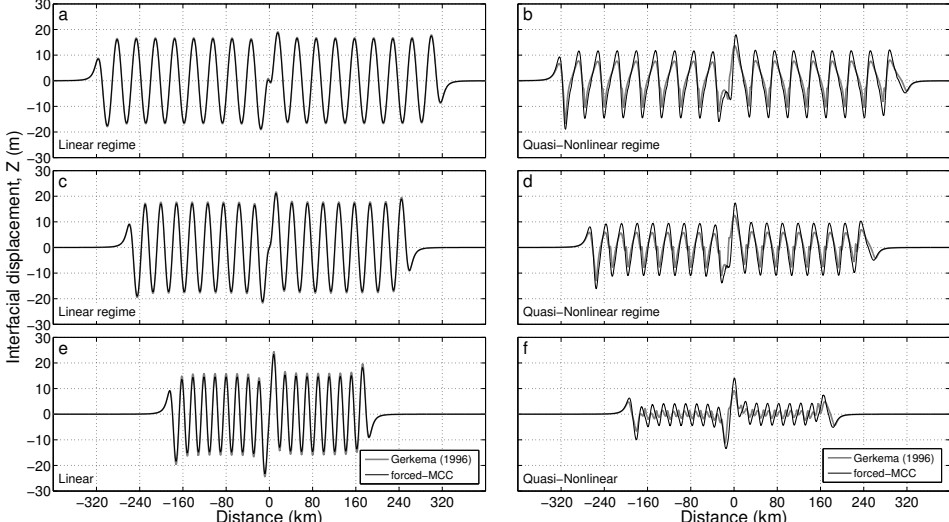

**Fig. 2.** Comparison of a linear (left panels) and quasi-nonlinear (right panels) internal tide generated via tidal flow over a sill (Gerkema, 1996) [grey line] with one generated by a horizontally oscillating sill (forced-MCC) [black line]. The upper and lower layer thickness are $H_1 = 30$ m and $H_2 = 70$ m, respectively. The (mimicked) tidal flow is fairly strong and equals $120$ cm s$^{-1}$ in all panels. The duration of the run is 9 tidal periods. Reduced gravity, $g'$, varies from top to bottom panels as: $g' = 0.03$ in (a, b), $g' = 0.02$ in (c, d) and $g' = 0.01$ in (e, f).

We notice that the linear runs with our forced-MCC model were actually done somewhat indirectly by taking the quasi-nonlinear version and reducing the forcing by a factor of 10, and afterwards enhancing the amplitude in the plots accordingly, since the quasi-nonlinear terms cannot be removed
explicitly in this model setting without also removing the forcing. By reducing the forcing, we effectively enter the linear regime.

Results over the top of the sill for various parameters, in Fig. 2, indicate a close correspondence between the two frames of reference. In line with this correspondence, we henceforth refer to the
speed of the oscillating topography as the "strength of the tidal flow".

It is only at relatively low values of $g'$ combined with a fairly strong tidal flow that we start to see significant deviations between the model settings (Fig. 2f). These deviations observed at low values of $g'$ vanish when forced with a tidal flow $\leqslant 80$ cm s$^{-1}$ (not shown).


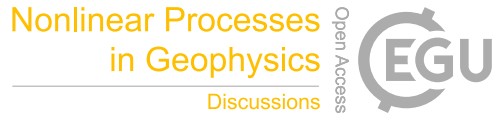

## 4 Numerical experiments

In this section, we use the forced-MCC-f set of equations derived in Sect. 2 to gain insight in the generation and evolution of the linear, quasi-nonlinear and fully nonlinear interfacial tides in ocean-like study cases. Special attention is given to conditions where interfacial waves reach limiting ampli-
tudes.

For convenience in comparing the different experiments, we configure the two-fluid layer system such that in all cases a total water depth of 100 m is considered ($H = 100$ in Eq. 54), with the upper layer being always thinner than the lower layer. Regarding the topography, we take different
heights but always with the same horizontal scale of approximately 20 km ($H_L = 10$ km in Eq. 54). Furthermore, the horizontal oscillation of the moving topography is always of semidiurnal frequency.

In Table 1 a list of the numerical experiments is provided, where the varying environmental conditions are defined following: the strength of the stratification, i. e. the reduced gravity, $g'$; the height
of the topography ($H_T$) relative to the lower layer thickness, referred to as the *topography ratio*, $\varphi_T$; the *two-layer thickness ratio*, $\gamma = H_1/H_2$; and, the latitude, $\phi$.

| Exp. | Regime (Fig.) | $g'$ (m s$^{-2}$) | $\varphi_T$ ($H_T$; in m) | $\gamma$ (H$_1$, H$_2$; in m) | $\phi$ (°) |
|---|---|---|---|---|---|
| A1$^{(*)}$ | **L** (3a-f), **QNL** (3a-f), **FNL** (6, 7, 8) | 0.03 | 0.57 (40) | 0.43 (30, 70) | 0 |
| A2 | **L** (3a,d), **QNL** (3a,d) | **0.02** | 0.57 (40) | 0.43 (30, 70) | 0 |
| A3 | **L** (3a,d), **QNL** (3a,d; 4) | **0.01** | 0.57 (40) | 0.43 (30, 70) | 0 |
| B1 | **L** (3b,e), **QNL** (3b,e), **FNL** (9, 11, 13a,b) | 0.03 | **0.50 (35)** | 0.43 (30, 70) | 0 |
| B2 | **L** (3b,e), **QNL** (3b,e) | 0.03 | **0.43 (30)** | 0.43 (30, 70) | 0 |
| C1 | **L** (3c,f), **QNL** (3c,f), **FNL** (10, 12, 13c,d) | 0.03 | 0.57 (40) | **0.33 (25, 75)** | 0 |
| C2$^{(*)}$ | **L** (3c,f), **QNL** (3c,f) | 0.03 | 0.57 (40) | **0.25 (20, 80)** | 0 |
| A1-f | **FNL** (14a,b) | 0.03 | 0.57 (40) | 0.43 (30, 70) | 15, 30, 45 |
| B1-f | **FNL** (14c,d) | 0.03 | **0.50 (35)** | 0.43 (30, 70) | 15, 30, 45 |
| C1-f | **FNL** (14e,f) | 0.03 | 0.57 (40) | **0.33 (25, 75)** | 15, 30, 45 |

**Table 1.** Summary of experiments. Capital letters L, QNL and FNL stand for Linear, Quasi-Nonlinear and Fully Nonlinear model configurations, respectively. Varying parameters are: the reduced gravity, $g'$ (m s$^{-2}$); the *topography ratio*, $\varphi_T$ ($H_T$ is indicated in meters); the *two-fluid layer thickness ratio*, $\gamma$ ($H_1$ and $H_2$ are indicated in meters); and, the latitude, $\phi$. The duration of the run in all experiments is 9 tidal periods. Experiments with an $^{(*)}$ were also performed under a linear configuration for varying conditions of $g'$ ranging from 0.005 to 0.05 m s$^{-2}$ (see Fig. 5).



### 4.1 Linear and quasi-nonlinear, hydrostatic interfacial tides

Regarding the rise of limiting amplitudes, we compare here the linear case (solid lines), where the
advective terms are absent, with the quasi-nonlinear case (dashed lines), where advective terms in-
volving interactions between the barotropic and baroclinic flows are included (however, interactions
between baroclinic fields, the genuinely nonlinear terms, are still absent).

We track systematically wave properties corresponding to the third leftward-propagating tide-generated
interfacial wave, counting from the front, after 9 tidal periods of forcing. By staying away from the
front, we avoid the transient effects affecting the waves that were generated at the start of the exper-
iment.

In panel (a), experiments A1, A2 and A3 account for the effect of varying the stratification via $g'$. In
panel (b), experiments A1, B1 and B2 account for the effect of varying *the topography ratio*, $\varphi_T$. In
panel (c), experiments A1, C1 and C2 account for the effect of varying *the two-fluid layer thickness
ratio*, $\gamma$. In each case, we consider a range of values of the amplitude of the barotropic tidal flow; by
increasing $a$ (dimensionless) in Eq. (47), we enhance the forcing via $U$.

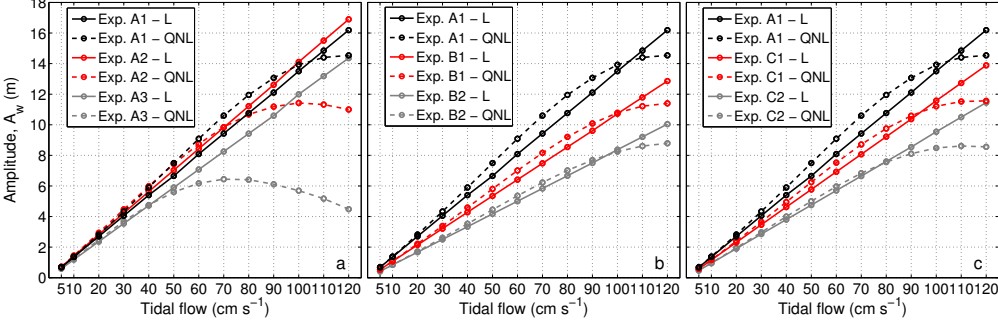

**Fig. 3.** Amplitude of the linear [solid] and quasi-nonlinear [dashed] interfacial tide, $A_w$ (m), vs. the strength of
the tidal flow (cm s$^{-1}$) for varying parameters: a) the strength of stratification, $g'$; b) the ratio of the height of
the topography to the lower layer thickness, $\varphi_T$; c) the two-layer thickness ratio, $\gamma$. See Table 1 for details on
the experiments. Capital letters L, and QNL stand for Linear and Quasi-Nonlinear model settings, respectively.

In the purely linear experiments (solid lines), the amplitude increases linearly with the amplitude of
the barotropic tidal flow, as one would expect. Naturally, for weak forcing the quasi-nonlinear cases
in Fig. 3 approach the linear ones; the advective terms then become very small. However, in the
quasi-nonlinear case (dashed lines), the interfacial waves exhibit a limiting amplitude in all exper-
iments (Fig. 3). Once the critical amplitude is attained, a further increase of the tidal forcing even
leads to a decrease of the wave amplitude. This behavior is further illustrated in Fig. 4 for experi-



ment A2, where a snapshot of the interfacial wave in the quasi-nonlinear regime is shown for various
forcing strengths. An increase of the forcing transforms the wave from sinusoidal to an asymmetric
shape, indicative of the generation of higher harmonics, but the amplitude becomes saturated.

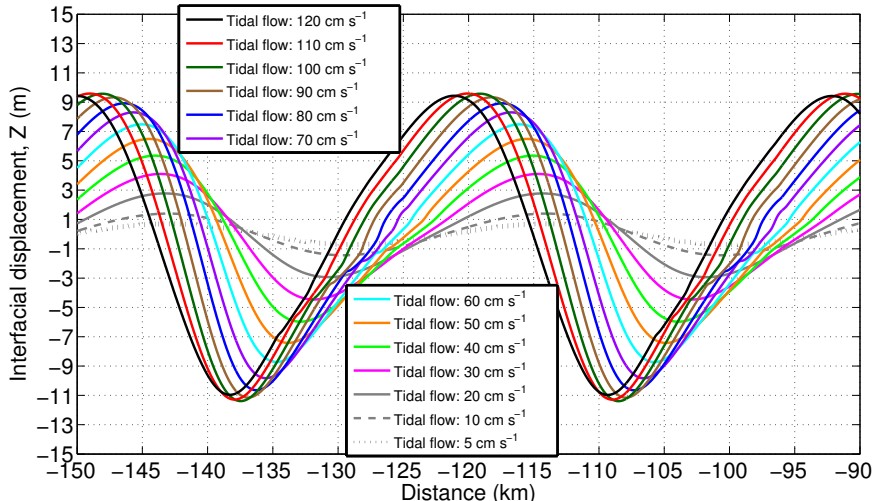

**Fig. 4.** Experiment A2 in Table 1. Snapshots of the quasi-nonlinear interfacial tide for various forcing strengths
(see legend) after 9 tidal periods of forcing.

For the quasi-nonlinear case in Fig. 3a, we observe that increasing the density jump through $g'$ gen-
erates larger interfacial waves which approach the linear regime, pointing to a weaker generation of
higher harmonics. The saturation and subsequent reduction of the quasi-nonlinear interfacial waves
becomes consequently more evident for a weaker stratified two-layer system (experiments A2 and
A3).


In relation to the increase of the *topography ratio* and the increase of the *two-fluid layer thickness
ratio*, both cases lead to larger linear and quasi-nonlinear interfacial waves, as shown in Fig. 3b,c.

The wavelength of the interfacial tide is in all cases independent of the amplitude of the barotropic
flow (not shown). Moreover, it follows from experiments A1, B1 and B2 that it is also independent
of the height of the topography. However, we find an increase in $L_w$ with increasing $g'$ or $\gamma$.

Interestingly, the linear cases in Fig. 3a also suggest that an increase of stratification generates larger
interfacial waves up to a certain threshold and, above it, the resulting wave is smaller; especially
noticeable for relatively strong tidal flows. Nevertheless, this feature is not observed in the quasi-





nonlinear case, where an increase of the density jump generates larger amplitudes along the three
cases of study. We explore this finding further in Fig. 5.

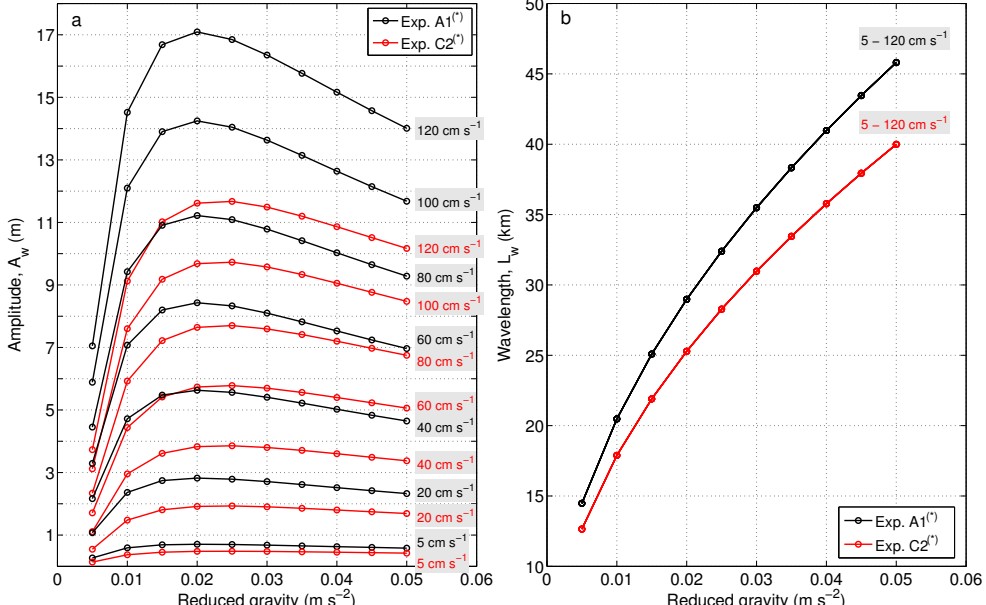

**Fig. 5.** Experiments A1*-L. (a) Amplitude, $A_w$ (m), of the linear interfacial wave vs. reduced gravity. (b)
Wavelength, $L_w$ (m), of the linear interfacial wave vs. reduced gravity. Labels indicate the strength of the tidal
flow for each experiment. See Table 1 for details on the experiments.

In Fig. 5a, the amplitude of the linear interfacial tide is shown for experiments A1$^{(*)}$ and C2$^{(*)}$,
where reduced gravity ranges from 0.005 to 0.05 m s$^{-2}$ and the forcing ranges from 5 to 120 cm s$^{-1}$.
In the literature, common values for $g'$ where solitary waves have been observed range from
0.007 m s$^{-2}$ in the Celtic Sea (Gerkema, 1996) to 0.027 m s$^{-2}$ over the Oregon continental shelf
(Stanton and Ostrovsky, 1998).

Results indicate that a range of optimal values of $g'$ exists and out of it the linear interfacial waves
decrease. This finding resembles to that of Gerkema (2001), where optimal values of $g'$ provide a
large interfacial amplitude (from generation by an impinging beam), while for very small or very
large values of $g'$ the resulting amplitude remains small. Furthermore, we find the optimal $g'$ be-
comes more apparent as the tidal flow increases, being almost absent for a weak tidal forcing. The
finding is consistent for experiments A1$^{(*)}$ and C2$^{(*)}$, which present different *two-fluid layer thick-
ness ratio*, $\gamma$ (see Table 1); although, broadly, this wave response is less apparent for a smaller $\gamma$
(experiment C2$^{(*)}$). In contrast, the wavelength of the linear interfacial waves presents no limiting



value within the range of study, although its growth is shown to decelerate as $g'$ increases for both A1$^{(*)}$ and C2$^{(*)}$ (Fig. 5b). As previously noted, varying the tidal forcing does not affect $L_w$.


Following the description above, the narrowest range of optimal $g'$ is found in experiment A1$^{(*)}$ between 0.015 and 0.03 m s$^{-2}$, when a fairly strong tidal flow of 120 cm s$^{-1}$ generates maximum amplitudes with less than 1 m of difference. For identical forcing strength, the range of optimal $g'$ broadens towards values up to 0.04 m s$^{-2}$ in experiment C2$^{(*)}$. If the wavelength of the interfacial

waves reaching their optimal amplitude in experiment A1$^{(*)}$ and C2$^{(*)}$ is compared with the width of the topographic sill, i. e. two times $H_L = 10$ km in Eq. (54), we find that interfacial waves approach their maximum amplitude for a ratio $L_w/(2 \times H_L)$ about 1.25–1.75 and 1.1–1.75, respectively (c. f. Figs. 5a and 5b), i. e. when their length scales are similar.

## 4.2 Weakly to fully nonlinear, weakly nonhydrostatic interfacial waves

In line with the results shown in Fig. 3, we select three different set-ups to investigate the fully nonlinear version of the forced-MCC equations, with all advective terms included. Accordingly, interfacial tides may disintegrate into strongly nonlinear solitons. Taking experiment A1 as the experiment of reference, B1 differs from A1 in that it has a lower $\varphi_T$, i. e. smaller height of the topography. C1 has

a lower $\gamma$ than A1, i.e., the maximum theoretical amplitude for strongly nonlinear waves is larger. Here, for identical strength of forcing, we test how nonlinear properties may differ depending on the environmental conditions.

### 4.2.1 Fully nonlinear interfacial waves: Experiment A1

Fig. 6a presents a spatial overview of tide-generated interfacial tides and solitons propagating leftward after 9 tidal periods of forcing with a tidal flow of 90 cm s$^{-1}$. In subsequent panels, a set of snapshots zooms in on different parts of the $x$-domain in panel (a) in order to capture different stages of the nonlinear disintegration of tide-generated interfacial waves.

Panel (b) shows the generation of a long internal tide of about 30 km wavelength. At a first stage, the internal tide splits up into two different groups of rank-ordered solitons: a train of depressions on the leading edge; and a train of elevations, after the former packet, with initially smaller amplitudes. At a later stage, panel (c), the largest elevations have reached the smaller depressions in the train and three leading solitons at the front present almost equal amplitudes about 29 m. Note that the

interfacial displacement of the soliton raises the interface between the two layers up to the domain of the upper layer $H_1$ (here the positive $y$-axis).



Previous solitary wave packets already propagating away from the generation area are shown in panels (d) and (e) and correspond to preceding disintegrated internal tides. The 'table-top' soliton

observed at the leading edge of every preceding internal tide emerged in all cases from the first of the three solitons described previously in panel (c).

As the leading soliton evolves and reaches its maximum amplitude, it also broadens, as predicted by soliton wave theory (Helfrich and Melville, 2006), in comparison with subsequent solitons of

smaller amplitude (Fig. 6d,e). The observed increase in the distance between the 'table-top' soliton and subsequent (smaller) solitons also indicates that, as expected from theory, the leading soliton moves (phase speed) faster than solitons in the tail.

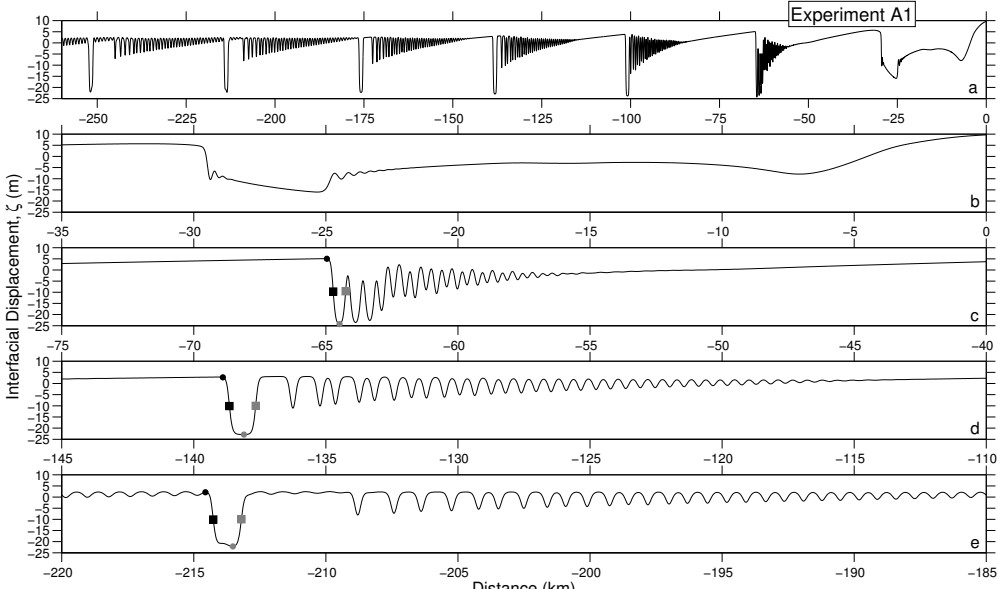

**Fig. 6.** Snapshots of experiment A1, forced with a tidal flow of 90 cm s$^{-1}$, after 9 tidal periods of the run. (a) Overview of leftward-propagating tide-generated interfacial waves. (b-e) Set of spatial zooms from (a) showing different stages of the nonlinear disintegration of tide-generated interfacial waves. As examples to show how the amplitude, $A_w^*$, and width, $L_w^*$, of tide-generated solitons are computed in this work, the points $Z_a$ (black dot), $Z_b$ (grey dot), $Z_c$ (black square) and $Z_d$ (grey square) are also indicated in (c-e) [see the text in Sect. 4.2.1 for details].

Because tide-generated solitons propagate through the evolving long interfacial tides, $z = 0$ cannot

be used as a reference level to compute the amplitude down to the trough of the solitary wave (see Fig. 1 and Fig. 6). Similarly, the soliton width cannot be measured along $z = 0$. A systematic criterion is required to adopt a suitable reference level which allows us to study the evolution of the soliton amplitude, $A_w^*$, and width, $L_w^*$.



Here we introduce the reference level $Z_a$, which for every leftward-propagating soliton locates where the first spatial derivative of the interfacial displacement, $Z$, approaches zero while being above a certain threshold. This grid-point indicates the location of the front of the leading soliton connecting with the tail of the preceding interfacial tide. Accordingly, the soliton amplitude, $A_w^*$, is defined as the vertical distance between $Z_a$ and the trough of the leading soliton, located at $Z_b$ (see, e. g., in Fig. 6c-e). The soliton width, $L_w^*$, is defined as the horizontal distance between $Z_c$ and $Z_d$, which locate half-way of the vertical distance spanning $A_w^*$ (see also, e. g., in Fig. 6c-e).

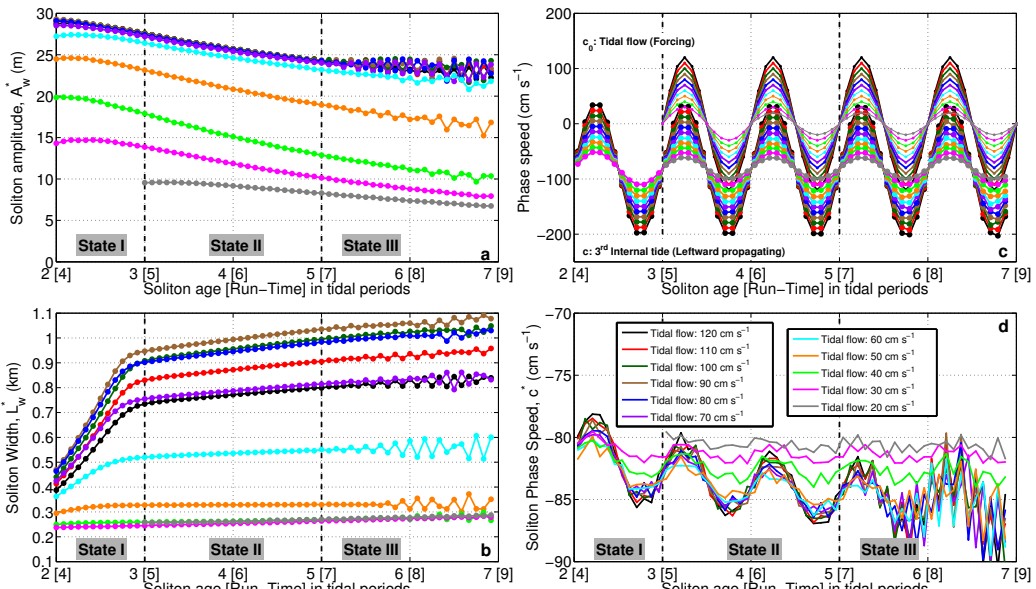

**Fig. 7.** Wave evolution of a tide-generated leading soliton tracked for experiment A1 under different forcing strengths (see legend). (a) Soliton amplitude (m) vs. soliton age (tidal periods). (b) Soliton width (km) vs. soliton age (tidal periods). (c) Phase speed (cm s$^{-1}$) of the tidal flow (thin lines) and phase speed of the leading soliton embedded in the internal tide (thick lines) vs. soliton age (tidal periods). (d) Soliton phase speed (cm s$^{-1}$) vs. soliton age (tidal periods). In all panels the lapse of time in the run, also in tidal periods, is provided in brackets along the $x$-axis.

The above criteria allow us to investigate the generation and evolution of the leading soliton towards a fully developed stage by tracking in space and time its wave properties. To this aim, we select the

third group of solitons and present the corresponding wave evolution in Fig. 7. Note that along the $x$-axis the soliton age is provided in tidal periods together with the corresponding run-time, indicated in brackets.




Contrary to what one might expect, the amplitude of the leading soliton decreases during its evolu-
tion (Fig. 7a). This can be ascribed to its tide-generated nature. At an early stage, the disintegration
of the interfacial tide leads at its front to a large depression, and this subsequently evolves to a mature
leading soliton propagating through the tail of the preceding internal tide (see Fig. 6c-e).

More elucidating is the soliton width evolution. We distinguish here three different states through
which the soliton evolves. These states are especially clear when the system is forced with a fairly
strong tidal flow ($\geqslant$50 cm s$^{-1}$). Then, the leading soliton emerges with a noticeable linear increase
of its width after 2 tidal periods of age (Fig. 7b; State I). During the subsequent two tidal periods,
the soliton width becomes nearly stationary, suggesting it has reached a fully developed state (State
II). Finally, after 5 tidal periods of age, the leading soliton overtakes the preceding internal tide,
interacting with small amplitude solitons along its tail and causing the observed oscillations in the
soliton width (State III).

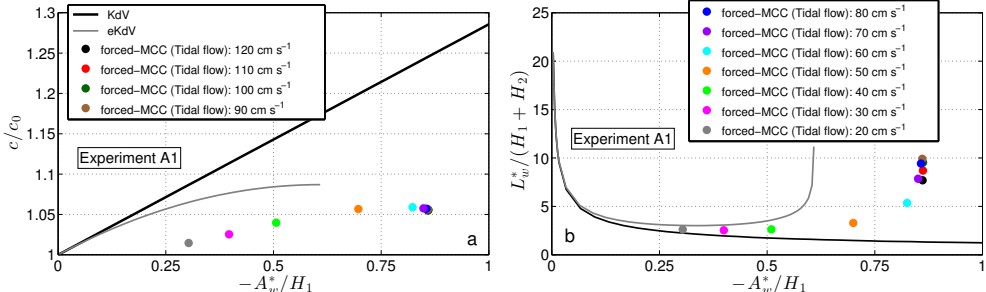

**Fig. 8.** Solitary wave solutions of the KdV (black line) and eKdV (grey line) theories compared to tide-generated
solitary waves derived from the forced-MCC equations (colored dots) for experiments A1. The tide-generated
solutions correspond to the mean mature leading soliton propagating in Fig. 7 (experiment A1) along its $4^{th}$
tidal period of age. The different colors indicate the strength of the tidal flow (see legend). (a) Nonlinear phase
speed scaled to linear long-wave phase speed for interfacial waves vs. soliton amplitude scaled to thickness of
the upper layer. (b) Soliton width scaled to total water depth vs. soliton amplitude scaled to thickness of the
upper layer. For comparison, the soliton width shown here for KdV and eKdV theories was computed following
the same procedure as for tide-generated solitons, i. e. we use points $Z_c$ and $Z_d$, as in Fig. 6c-e, applied to the
soliton theoretical solutions.

The soliton reaching its maximum amplitude broadens up to a certain maximum width, attaining the
'table-top' shape as the tidal flow is increased. However, above a certain threshold of forcing, the
soliton width decreases while keeping its maximum amplitude. We observe this when an increase of
the tidal flow beyond 90 cm s$^{-1}$ (maximum $A_w^*$ and $L_w^*$ along the brown line in Fig. 7a,b) does not
generate larger solitons but narrower, which are consequently not 'table-top' shaped. Because of the
nature of classical eKdV and MCC theory in absence of forcing, this feature could not rise before





and it does indicate that limiting factors related to the forcing may be acting.


The nearly constant soliton width shown for tidal flows weaker than 50 cm s$^{-1}$ is suggestive of the weakly nonlinear nature of the emerging solitons (Fig. 7b). In agreement with this, a sharp difference is also noted regarding the evolution of the soliton phase speed when the two-fluid system is forced with either a relatively weak/moderate tidal flow ($<$50 cm s$^{-1}$) or a strong tidal flow ($\geqslant$50 cm s$^{-1}$)

(Fig. 7d). Results show that tide-generated strongly nonlinear solitons present nearly the same phase speed in all cases (tidal forcing $\geqslant$50 cm s$^{-1}$), increasing from 80 cm s$^{-1}$ to 90 cm s$^{-a}$ as the soliton evolves towards its $7^{th}$ tidal period of age. It is also worth noting that the increase of the soliton phase speed exhibits an oscillation amplitude of about 5% of its value, and that this oscillation appears in phase with the tidal flow (Fig. 7c). Alternatively, tide-generated weakly nonlinear solitons

(tidal forcing $<$50 cm s$^{-1}$) present a nearly constant phase speed with a smaller amplitude of oscillation. Furthermore, within this regime, the soliton phase speed depends on the strength of the tidal flow.

Lastly, Fig. 8 compares the solitary wave solutions of the KdV and eKdV theories

(Kakutani and Yamasaki, 1978; Ostrovsky and Stepanyants, 1989; Helfrich and Melville, 2006; Gerkema and Zimmerman, 2008) with the wave properties of leading tide-generated solitons derived from the forced-MCC equations for experiment A1.

Generally, the relationship between the soliton phase speed and the soliton amplitude follows a sim-

ilar curve to that predicted from the eKdV theory, although experimental phase speed values are weaker in all cases (Fig. 8a). As regards to the relationship between the soliton width and amplitude, tide-generated solitons follow a parallel curve to that predicted by eKdV, with similar width but larger maximum amplitude, reaching the 'table-top' shape at its maximum, as expected (Fig. 8b).

**4.2.2 Strongly and weakly nonlinear interfacial waves: Experiments B1 and C1**

In this section we perform and discuss analogous analyses to those presented for experiment A1. Thus, Figs. 9a and 10a show an overview of tide-generated interfacial waves propagating leftward after 9 tidal periods of forcing with a tidal flow of 100 cm s$^{-1}$ for experiments B1 and C1, respectively. In subsequent panels, a set of snapshots zooms in on different parts of the $x$-domain in (a) in

order to capture different stages of the nonlinear disintegration of tide-generated interfacial waves. Note that here the selected strength of the tidal flow is 100 cm s$^{-1}$, by contrast with Fig. 6 (experiment A1) where this is 90 cm s$^{-1}$. We have selected in each case the forcing strength leading to the maximum experimental soliton amplitude and width.



Different from experiment A1, here the internal tides do not split up into two different groups of
solitons (Figs. 9b and 10b), but disintegrate into solitary wave packets of rank-ordered depressions,
with the leading soliton reaching a maximum experimental amplitude about 27 m in panel (c) of
both experiments, B1 and C1 (Figs. 9 and 10).

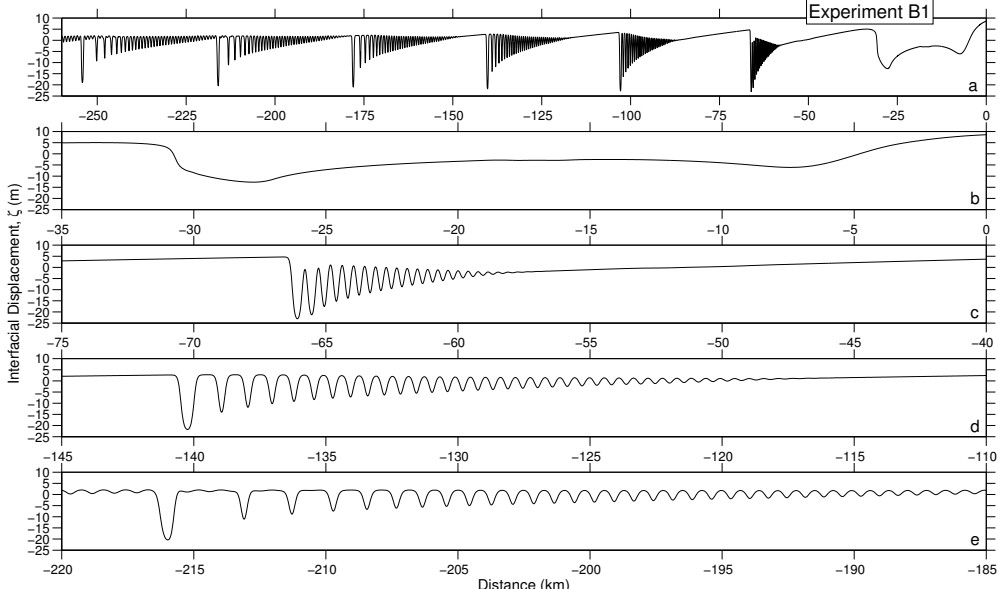

**Fig. 9.** Snapshots of experiment B1, forced with a tidal flow of 100 cm s$^{-1}$, after 9 tidal periods of the run. (a)
Overview of leftward-propagating tide-generated interfacial waves. (b-e) Set of spatial zooms from (a) showing
different stages of the nonlinear disintegration of tide-generated interfacial waves.

Preceding disintegrated internal tides, i. e. more at the front of the wave train, are shown for experi-
ments B1 and C1 in Figs. 9 d,e and 10 d,e. Remarkably, the 'table-top' solitons at the leading edge
of the long-life internal tides in experiment A1 (Fig. 6d,e) are absent for experiments B1 and C1.
The lower height of the topography for experiment B1, and the decrease of the upper layer thickness
for experiment C1 (a larger amplitude is required to obtain 'table-top' solitons) seems to be the main
cause.

Although starting with leading solitons of relatively similar amplitude, experiment C1 exhibits sig-
nificantly smaller and narrower leading solitons than A1 (c. f., Figs. 6d,e and 10d,e). This suggests
that dispersive effects might overcome nonlinearities more noticeably along the solitons when the
upper layer is thinner.

In Figs. 11 and 12, the generation and evolution of an emerging leading soliton is tracked for exper-





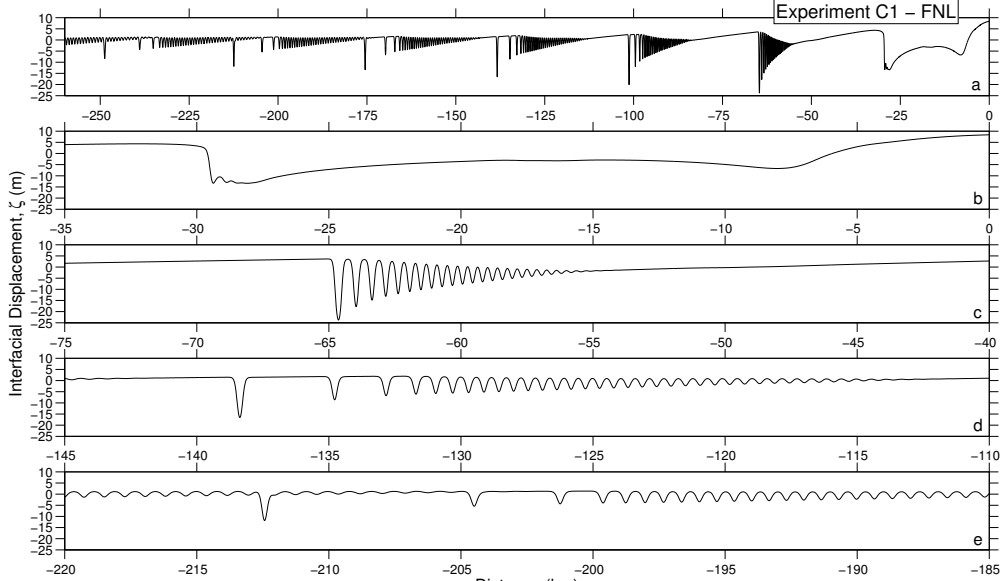

**Fig. 10.** Snapshots of experiment C1, forced with a tidal flow of 100 cm s$^{-1}$, after 9 tidal periods of the run. (a) Overview of leftward-propagating tide-generated interfacial waves. (b-e) Set of spatial zooms from (a) showing different stages of the nonlinear disintegration of tide-generated interfacial waves.

iments B1 and C1, respectively, as it was done for experiment A1 in Fig. 7. The tracked soliton is also here the third tide-generated leftward-propagating leading soliton, counting from the front.


A striking feature emerges for leading solitons in experiments B1 and C1, which do not exhibit a 'table-top' form, but nevertheless attain a limiting amplitude that cannot be exceeded by further increasing the barotropic tidal flow (Figs. 11a and 12a). They reach this experimental limiting amplitude when the tidal forcing is 100 cm s$^{-1}$, while in experiment A1 it is reached for a weaker

tidal flow about 90 cm s$^{-1}$. Importantly, the soliton width decreases for experiments B1 and C1 when leading solitons attain their empirical limiting amplitude and the tidal forcing is increased further. This decrease of the soliton width resembles to that previously noted for experiment A1 when the tidal flow is increased beyond the forcing strength leading to 'table-top' solitons. Additionally, leading solitons in experiments B1 and C1 also decrease their amplitude under the discussed circum-

stance. This is observed for tidal flows higher than 100 cm s$^{-1}$, which generate solitons of smaller amplitudes, more noticeably in experiment C1 than in B1 (c. f., Fig. 11a and 12a).

Following the above descriptions, we find that tidally generated solitons are subjected to limiting amplitudes, even under weakly nonlinear conditions, due to the generation of higher harmonics as

the tidal forcing increases, which already limit the growth of the initial internal tide (see Sect. 4.1).



As in experiment A1, two groups of leading solitons emerge for experiment B1. The strongly non-linear solitons, which exhibit an initial linear growth of their width and a nearly similar, and oscillating, increase of their phase speed. And, the weakly nonlinear solitons, which exhibit a nearly
constant width and phase speed (Figs. 11b,d). Alternatively, for experiment C1, all the leading solitons present a nearly constant width, suggesting the prevalence of weakly nonlinearities (Figs. 12b). Regarding the soliton amplitude and phase speed, however, we also observe for experiment C1 two distinguishable groups: on the one hand, larger solitons which present nearly the same oscillating phase speed with no trend; and, on the other hand, smaller solitons which present a nearly constant
phase speed that depends on the strength of the tidal flow (Figs. 12a,d). As it occurred for experiment A1, the oscillating phase speed of the leading solitons in experiments B1 and C1 also appears in phase with the tidal flow (Figs. 11c and 12c).

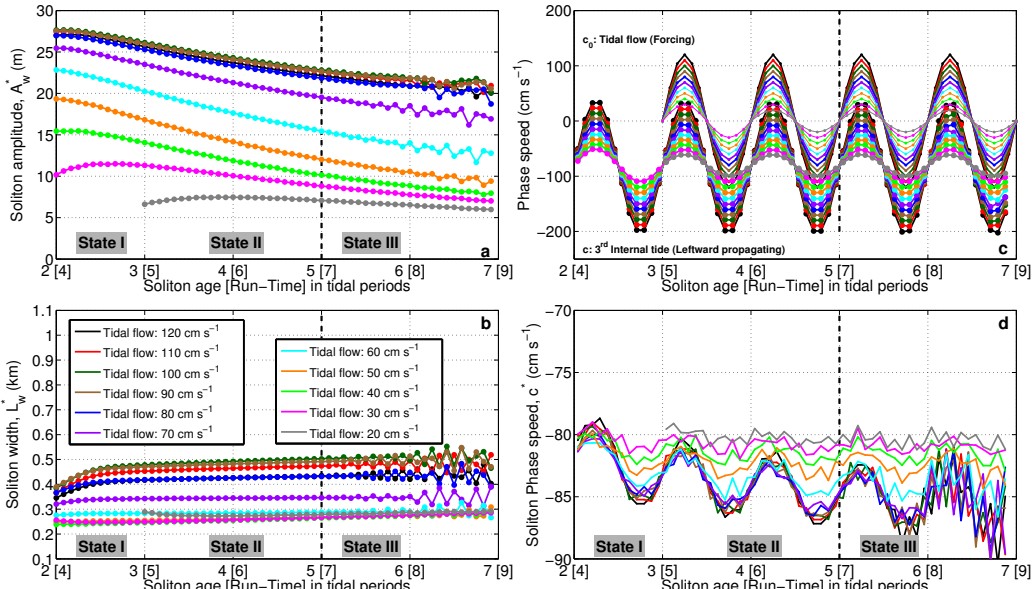

**Fig. 11.** Wave evolution of a tide-generated leading soliton tracked for experiment B1 under different forcing strengths (see legend). (a) Soliton amplitude (m) vs. soliton age (tidal periods). (b) Soliton width (km) vs. soliton age (tidal periods). (c) Phase speed (cm s$^{-1}$) of the tidal flow (thin lines) and phase speed of the leading soliton embedded in the internal tide (thick lines) vs. soliton age (tidal periods). (d) Soliton phase speed (cm s$^{-1}$) vs. soliton age (tidal periods). In all panels the lapse of time in the run, also in tidal periods, is provided in brackets along the $x$-axis.

When compared to the solitary wave solutions of the KdV and eKdV theories, numerical results
from using the forced-MCC equations show that generation of strongly and weakly nonlinear soli-




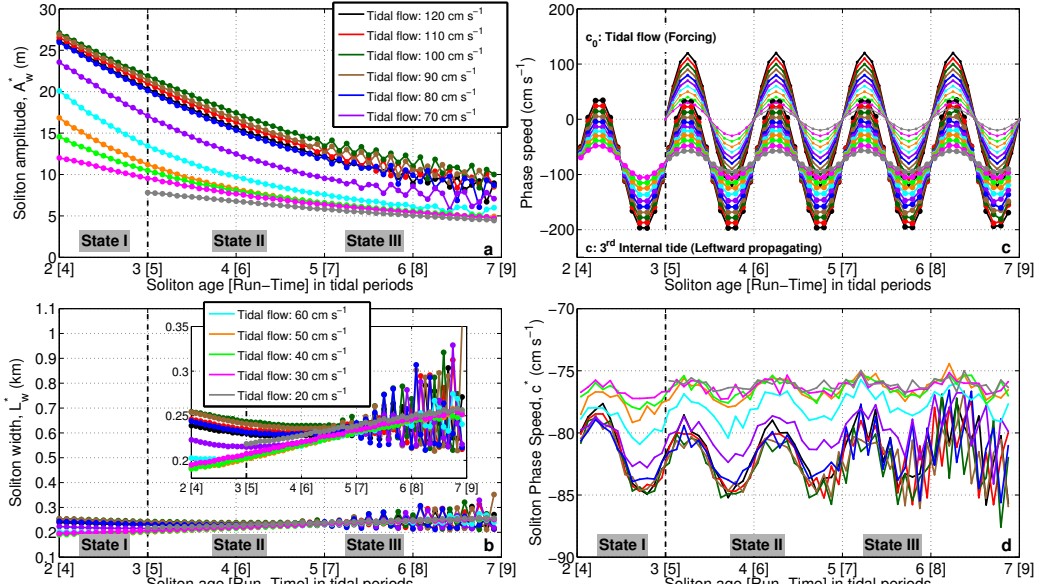

**Fig. 12.** Same as Figure 11 but for experiment C1.

tons occur for experiments B1 and C1 (Fig. 13).

In experiment B1, the relationship between the soliton phase speed and the soliton amplitude follows a similar curve to that predicted from the eKdV theory (Fig. 13a), although experimental phase
speed values are weaker in all cases, as it also occurred for experiment A1 (see Fig. 8a).

Conversely, tide-generated solitons from experiment C1 follow more closely a linear trend which resembles that predicted by KdV theory, also showing in all cases weaker experimental phase speed values (Fig. 13c).


Interestingly, while not attaining a 'table-top' form, nonlinear solitons in experiments B1 and C1 also present a limiting amplitude; even though in experiment B1, the largest solitons start to slightly broaden and present amplitudes larger than that predicted by eKdV theory. Furthermore, it is also evident in Fig. 13b,d that the largest solitons decrease their amplitude if the tidal flow increases
above a certain value, as previously noted from Figs. 11 and 12. These results further suggest that it is the saturation of the underlying quasi-nonlinear internal tide affected by higher harmonics (see Fig. 3), which eventually limits the growth of tide-generated solitons.





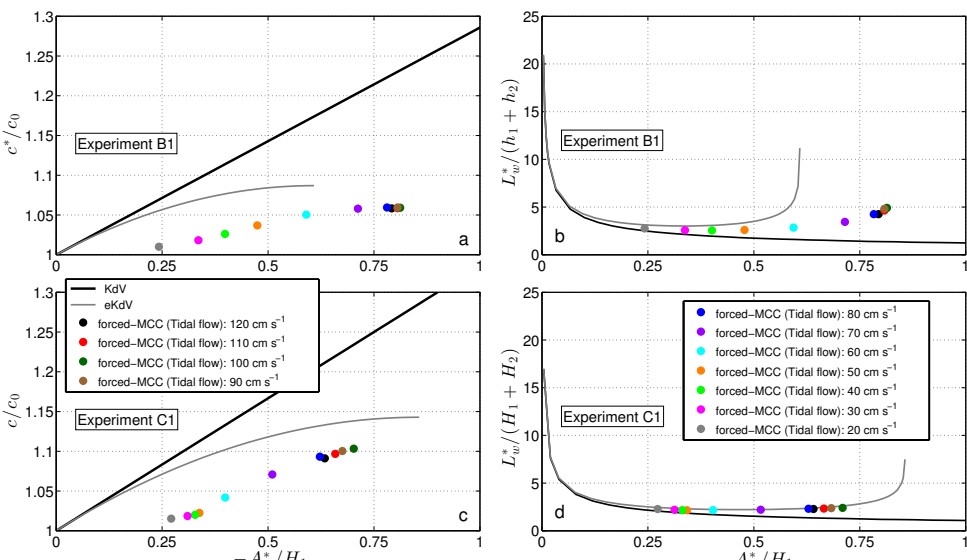

**Fig. 13.** Solitary wave solutions of the KdV (black line) and eKdV (grey line) theories compared to tide-generated solitary waves derived from the forced-MCC equations (colored dots) for experiments B1 (top row) and C1 (bottom row). The tide-generated solutions correspond to the mean mature leading soliton propagating in Figs. 11 (experiment B1) and 12 (experiment C1) along its $4^{th}$ tidal period of age. The different colors indicate the strength of the tidal flow (see legend). (a,c) Nonlinear phase speed scaled to linear long-wave phase speed for interfacial waves vs. soliton amplitude scaled to thickness of the upper layer. (b,d) Soliton width scaled to total water depth vs. soliton amplitude scaled to thickness of the upper layer. For comparison, the soliton width shown here for KdV and eKdV theories was computed following the same procedure as for tide-generated solitons, i. e. we use points $Z_c$ and $Z_d$, as in Fig. 6c-e, applied to the soliton theoretical solutions.

### 4.3 Effects of the Earth's rotation

Finally, we investigate rotational effects acting as a dispersive mechanism on the wave evolution of tidally-generated solitons derived from the forced-MCC-f equations presented in this work.

In Fig. 14, numerical solutions are shown for experiments A1-f, B1-f and C1-f (see Table 1 for details). The different colored lines show how comparatively rotation affects the nonlinear interfacial

tides and solitons. Rotation becomes gradually stronger as follows: rotationless (black line), $\theta$=15° (grey line), $\theta$=30° (blue line) and $\theta$=45° (red line).

In agreement with previous studies, we observe in all panels that an increase of the latitude leads to larger dispersive effects due to Coriolis dispersion (Gerkema and Zimmerman, 1995; Gerkema,

1996). Also, the leading solitons are shown to travel faster as rotation becomes stronger due to rota-



tion increases the phase speed of a wave.

Importantly, an unexpected feature must be noted. At relatively low latitudes (c. f., numerical so-
lutions for the rotationless case and $\theta$=15°), rotational effects seem to enhance the growth of the
leading solitons up to reaching a 'table-top' form. This is especially noticeable for experiments B1
and C1, panels (d) and (f), respectively, for which the rotationless cases do not lead to 'table-top'
solitons.

These results indicate that a moderate increase of rotation contributes to decrease the effect of lim-
iting factors preventing the growth of leading solitons as 'table-top' solitary waves. In previous
sections we have shown that higher harmonics generated by strong tidal flows cause a saturation
in the quasi-nonlinear tide by which solitons emerge, and that this factor limits the growth of the
leading soliton. Accordingly, we suggest that at relatively low latitudes, the saturation of the quasi-
nonlinear interfacial tide may vanish as the higher harmonics disperse due to Coriolis dispersion. As
a consequence, leading solitons appear attaining a 'table-top' form under forcing conditions which
do not generate such solitary waves in the rotationless cases.

At higher latitudes ($\theta$=45°), the dispersive effect of the Coriolis force becomes stronger than non-
linearities, thus preventing the nonlinear interfacial tide from disintegrating into strongly nonlinear
solitons, as expected.

## 5   Summary and conclusions

We investigate limiting amplitudes of tides and solitons using a generalization of the fully nonlinear
MCC equations (Miyata, 1985, 1988; Choi and Camassa, 1999), extended here with forcing terms
and Coriolis effects (forced-MCC-f). The focus is on the effects of adding a forcing, which repre-
sents a novelty in the existing literature and provides a closer view to an ocean-like scenario. The
mechanism for interfacial tide generation is represented by a horizontally oscillating sill, mimicking
a barotropic tidal flow over topography. Solitons are generated by a disintegration of the interfacial
tide.


Numerical solutions show that strongly nonlinear tide-generated solitons attain in some cases a lim-
iting table-shaped form, in agreement with classical soliton theory. Nevertheless, results also reveal
that tide-generated solitons may be also limited by saturation of the underlying quasi-nonlinear in-
ternal tide. The decisive factor is then the generation of higher harmonics, which are enhanced with
increasing forcing and are here demonstrated to limit the growth of the initial internal tide before



disintegration into solitons. This effect seems to have passed unnoticed in previous studies, but turns out to be be a key factor in the subsequent generation of solitons. It implies that under strongly nonlinear conditions, amplitudes of solitons may be limited before attaining a table-shaped form. Interestingly, the increase of the tidal generation of 'table-top' solitons causes a progressive reduc-

tion of the soliton, narrowing and, in some cases, even decreasing its amplitude. Importantly, we find that tidally generated solitons are subjected to limiting amplitudes even under weakly nonlinear conditions, contrary to predictions by classical KdV theory alone. In the weakly nonlinear regime too, the increase of the tidal forcing above a certain threshold causes a progressive reduction of the tide-generated soliton. The upshot is that increasing the tidal forcing above a certain strength does

not lead to larger solitons but, counterintuitively, to smaller ones.

Numerical solutions also show that the time-scale for a tide-generated leading soliton to reach a mature stage is shorter for weakly than for strongly nonlinear solitons. The former adopt a final form relatively fast after its emergence, smoothly evolving to a mature soliton since its second tidal period

of existence. Alternatively, strongly nonlinear solitons adopt a final form after a noticeable transition which occurs after its second tidal period of existence.

Another departure from classical eKdV theory is that strongly nonlinear tide-generated solitons exhibit larger maximum amplitudes than predicted, while soliton phase speeds are smaller for both

weakly and strongly nonlinear solitons. We suggest that the larger amplitude might be caused, to some extent, by the nature of the tide-generated soliton, which rides on an interfacial tide that is lifted above the interface level at rest, hence allowing for larger maximum amplitudes. The question remains unresolved about the factors causing the slower than predicted soliton phase speed.

In relation to the limiting amplitudes in rotational cases, numerical results from the forced-MCC-f equations suggest that a moderate increase of rotation contributes to decrease the effect of limiting factors preventing the growth of leading solitons as 'table-top' solitary waves. We suggest that at relatively low latitudes, the saturation of the quasi-nonlinear interfacial tide may vanish as the higher harmonics disperse due to Coriolis dispersion. Consequently, leading solitons can attain a 'table-

top' form under forcing conditions that would not generate such solitary waves in the rotationless case. As expected, if rotation becomes stronger, the dispersive effect of the Coriolis force becomes stronger than nonlinearities and the nonlinear interfacial tide is prevented from disintegrating into strongly nonlinear solitons.





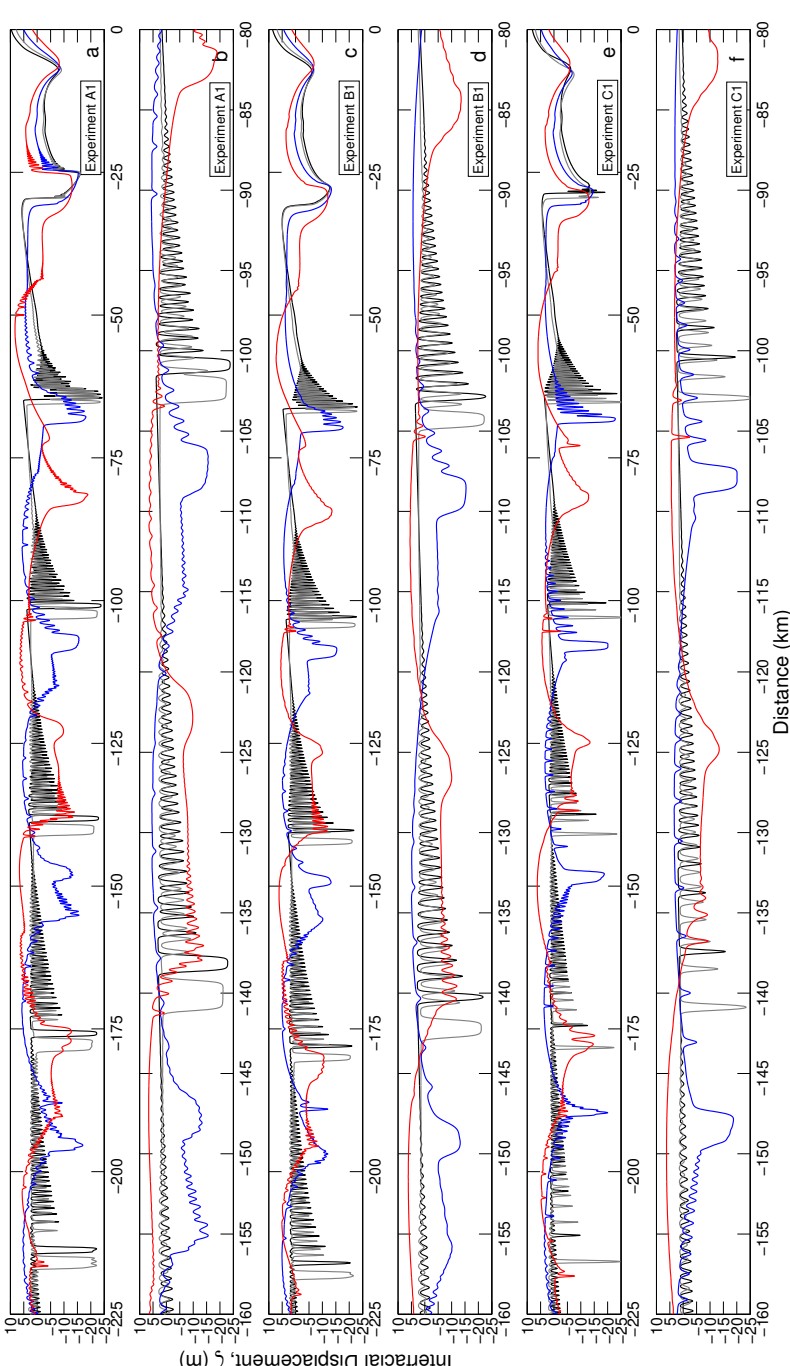

**Fig. 14.** Snapshots of experiment A1 (tidal flow: 90 cm s$^{-1}$), B1 (tidal flow: 100 cm s$^{-1}$) and C1 (tidal flow: 90 cm s$^{-1}$), indicated with labels in each panel, after 9 tidal periods of the run. (a,c,e) Overview of leftward-propagating tide-generated interfacial waves. (b,d,f) Spatial zoom from the corresponding overview. In all panels the rotationless case is shown as a black line, while rotating cases are shown, respectively, for $\theta=15°$ (grey line), $\theta=30°$ (blue line) and $\theta=45°$ (red line).





*Acknowledgements.* Financial support for this research was provided by the Spanish government (Ministerio de Ciencia e Innovación) through a Ph.D. grant (FPU) awarded to the first author (AP2007-02307).

**Appendix A**

**Numerical strategy**

We define a grid in time and space for discretization of the various derivatives of the system. Then,

$$t_n = n\Delta t \qquad and \qquad x_j = j\Delta x$$

are introduced for integer values of $n$ (time-step) and $j$ (spatial-step), where $\Delta t$ and $\Delta x$ are the magnitude of the steps. Time and spatial dependent variables are described as, e.g. $y(t_n, x_j)$, at any time and position. Thus, $y_j^n$ means the value of the variable $y$ at the current time and spatial-step, $n$ and $j$, respectively. And, consequently, $n+1$ represents the '*next time-step*', and so $n-1$ the '*previous*

*time-step*', what applies analogously for $j$ in the space grid.

The various derivatives in the model are discretized with centered difference approximations (Durran, 1999) as follows

$$y_t(t_n, x_j) \cong \frac{y_j^{n+1} - y_j^n}{\Delta t}, \tag{A1}$$

$$y_x(t_n, x_j) \cong \frac{y_{j+1}^n - y_j^n}{\Delta t}, \tag{A2}$$

$$y_{xx}(t_n, x_j) \cong \frac{y_{j+1}^n - 2y_j^n + y_{j-1}^n}{(\Delta x)^2}, \tag{A3}$$


$$y_{xt}(t_n, x_j) \cong \frac{y_{j+1}^{n+1} - y_{j+1}^n - (y_{j-1}^{n+1} - y_{j-1}^n)}{2\Delta x \Delta t}, \tag{A4}$$

$$y_{xxt}(t_n, x_j) \cong \frac{y_{j+1}^{n+1} - y_{j+1}^n - 2(y_j^{n+1} - y_j^n) + (y_{j-1}^{n+1} - y_{j-1}^n)}{(\Delta x)^2 \Delta t}. \tag{A5}$$

$$\tag{A6}$$

Initiatily the system is at rest with mean horizontal velocities, $\bar{u}_i$ and $\bar{v}_i$, and displacement of the interface, $\zeta$, being all zero at the first two time levels $(n-1, n)$, what represent the initialization fields. The thickness of the upper, $h_1$, and lower layer, $h_2$, together with the topography, $h(X)$, draw the scenario where the two-layer system runs. At the next time-step $(n+1)$, we start to move the topography to the right creating the effect of a tidal motion flowing to the left. For given $U$, i. e.

scaled velocity of moving topography (Eq. (47)), and time-step, the excursion of the topography is





a known quantity which is used to shift (first, second and third) spatial derivatives of $h(X)$ at every new time-step.

The time derivatives of the $\bar{v}_i$–momentum and continuity equations (51), (52) and (53) are solved

numerically using the third-order Adams-Bashforth approximation (Durran, 1999), for which $\bar{v_1}$, $\bar{v_2}$ and $\zeta$ at the next time-step $(n+1)$, and at all $j$ positions, are determined in terms of the known quantities at the previous two time-steps $(n-1, n)$.

However, solving numerically $\bar{u}_1$ from Eq. (49) is not straightforward as we deal with three different

time derivatives of $\bar{u}_1$ accompanied with space-time-dependent coefficients. Thus, after collecting the various time derivatives involving $\bar{u}_1$ on one side and remaining terms on the other side, the horizontal momentum equation of $\bar{u}_1$ evolves to a generic expression in the form of

$$a\,\bar{u}_{1,t} + b\,\bar{u}_{1,xt} + c\,\bar{u}_{1,xxt} = Y(t_n, x_j) \tag{A7}$$

where $a$, $b$ and $c$ collect spatial derivatives of space-time dependent variables ($\zeta(x,t)$ and $h(x,t)$);

and, $Y(t_n, x_j)$ represents a collection of known quantities whose values may be dependent on time and/or space. In the remainder, we describe the numerical strategy we follow to solve this problem.space-time-dependent partial differential equations. If now we operate the time derivative as a common factor in the left-hand side, the result leads to

$$(a\,\bar{u}_1 + b\,\bar{u}_{1,x} + c\,\bar{u}_{1,xx})_t = Y(t_n, x_j) + (a_t\,\bar{u}_1 + b_t\,\bar{u}_{1,x} + c_t\,\bar{u}_{1,xx}) \tag{A8}$$

what helps us to introduce a new variable, $\bar{U}_1$, which groups coefficients $a$, $b$, $c$ and time derivatives of $\bar{u}_1$ and turns our problem into a numerically solvable expression in the form of

$$\overline{U_1}_{,t} = Y(t_n, x_j) + (a_t\,\bar{u}_1 + b_t\,\bar{u}_{1,x} + c_t\,\bar{u}_{1,xx}) \tag{A9}$$

It is important to recall here that $Y(t_n, x_j)$ and the spatial derivatives of $\bar{u}_1$ are both evaluated at the current time-step $(n)$; and, the time derivatives of $a$, $b$ and $c$, which involve values of $\zeta$ at the current

$(n)$ and new time-step $(n+1)$, have been previously achivied with Eq. (53) via Adams-Bashforth approximation. This allows to rewrite the above expression as

$$\overline{U_1}_{,t} = R(t_n, x_j) \tag{A10}$$

by grouping all known quantities on the right-hand side under the variable $R(t_n, x_j)$. Next we need to discretize the time derivative of $\overline{U_1}$ but before doing that, we work out and discretize its spatial

derivatives using Eqs. (A2) and (A3), what results in

$$\overline{U_1} = \left(a_j - \frac{2\,c_j}{2\Delta x}\right)\bar{u}_{1j} + \left(\frac{-b_j}{2\Delta x} + \frac{c_j}{(\Delta x)^2}\right)\bar{u}_{1j-1} + \left(\frac{b_j}{2\Delta x} - \frac{c_j}{(\Delta x)^2}\right)\bar{u}_{1j+1}$$

which we rewrite by introducing factors $d$, $e$ and $f$ as follows

$$\overline{U_1}_j = d_j\,\overline{u_1}_j + e_j\,\overline{u_1}_{j-1} + f_j\,\overline{u_1}_{j+1}\,. \tag{A11}$$



If we now discretize the time derivative of $\overline{U_1}$ and apply Adams-Bashftorth, we obtain a numerically solvable expression for $\overline{U_1}$ at the next time step, which reads

$$\overline{U_1}_j^{n+1} = \overline{U_1}_j^{n} + \frac{\Delta t}{12}\left(23R_j^n - 16R_j^{n-1} + 5R_j^{n-2}\right), \qquad (A12)$$

where $\overline{U_1}_j^{n+1}$ actually includes

$$\overline{U_1}_j^{n+1} = d_j^{n+1}\,\bar{u_1}_j^{n+1} + e_j^{n+1}\,\bar{u_1}_{j-1}^{n+1} + f_j^{n+1}\,\bar{u_1}_{j+1}^{n+1}. \qquad (A13)$$

To close our system we still need to obtain $\bar{u_1}_j^{n+1}$ for all $j$ terms. To that end, the equation above is more complicated to solve and gives rise to implicit equations, as we have not only the unknown $\bar{u_1}_j^{n+1}$, but also $\bar{u_1}_{j-1}^{n+1}$ and $\bar{u_1}_{j+1}^{n+1}$, which come from the mixed second and third derivatives of $u_1$ in Eq. (A7). However, this is a well-known problem that can be solved using the tridiagonal matrix algorithm (TDMA), also known as the Thomas algorithm (Logan, 1987).

Following the numerical strategy described above, the model resolution is closed for every new time level $n+1$ and the model equations can be solved successfully.

The choice of the space-time steps $\Delta t$ and $\Delta x$ is based on two main requirements. Firstly, the resolution in $x$ ($\Delta x$) must be sufficiently fine to resolve third-derivative terms and ensure that any short, solitary-like waves are properly resolved. Nevertheless, dealing with equivalent equations to Miyata (1988) and Choi and Camassa (1999), as we do in our model, Kelvin-Helmholtz instabilities are not filtered out. In this regard, Jo and Choi (2002) found that solitary waves of sufficient amplitude could be unstable at high wave numbers to Kelvin-Helmholtz instability. Thus, if the grid resolution is too fine, unstable short waves will emerge near the wave crest and ultimately overwhelm the calculations and explode numerically (Jo and Choi, 2002; Helfrich and Melville, 2006; Helfrich and Grimshaw, 2008). In some cases, the instability can be controlled by filtering out wavenumbers above a threshold (W. Choi 2007, personal communication cited in Helfrich and Grimshaw (2008)). For our numerical experiments we consider a $\Delta x$ course enough to prevent the problem. A second condition follows from the requirement of stability. Then, for a given spatial step one may take the Courant-Friedrichs-Lewy condition for the linearized equations as an indication of the required time step. The criterion implies that $\Delta x / \Delta t$ should be larger than the phase speed of the wave; taking special care where the advection by the barotropic tidal flow (here mimicked with the moving topography) should be added to the phase speed to apply the criterion properly (Gerkema, 1994).

For the simulations we present, it was not needed to filter out wavenumbers above a threshold to control Kelvin-Helmholtz instabilities as we designed the space-time grid to avoid this problem following previous conditions. However, in some cases, specially in the simulations where the forcing was fairly strong, an additional trick was needed to retain stability around the generation area (Gerkema, 1994). In those cases averages were taken in the vicinity of the top of the ridge (around


the steepest part of the topography), where the instabilities arised. At one particular point $(x_j, t_n)$ in space-time, new values of $\bar{u}_i$, $\bar{v}_i$ and $\zeta$ were calculated by taking the average of the old values at $x_{j-1}$, $x_j$ and $x_{j+1}$, and subsequently in time between $t_n$ and $t_{n-1}$. The disturbance provoked by this procedure was tested and found to be a minor effect only, as it was only applied over the closest region to the top of the topography.


### Appendix B

### Numerically solvable model equations

In Appendix A, the numerical scheme used to solve the model is explained using a generic expression (A7) for the $\overline{u_i}$ horizontal momentum equation (49). Here we present the full set of nondimensional

forced-equations actually used for the numerical solving of the model. The procedure to that end is as follows.

Firstly, all terms of the $\overline{u_i}$ horizontal momentum equation (49) are worked out and grouped according to their physical effects (i. e. *linear*, *nonlinear* and *dispersive effects* from the upper and lower layer,

and from topography), leaving unkown quantities involving time derivatives of $\overline{u_1}$ on the left-hand side. The resulting expression (31) resembles (A7), where coefficients $a$, $b$ and $c$ involve derivatives of space-time dependent variables and $Y(t_n, x_j)$ is represented here by the sum of all terms on the right-hand side,

$$a\,\bar{u}_{1,t} + b\,\bar{u}_{1,xt} + c\,\bar{u}_{1,xxt} = linear + nonlinear + dispersive_1 + dispersive_2 + dispersive_{topo}$$

(31)

$$+\delta_2\Big[(\eta_2 h_x - \eta_2\zeta_x)\phi_x - \frac{\eta_2^2}{3}\phi_{xx} + \phi\big(\frac{\eta_2}{2}h_{xx} + \zeta_x h_x\big)\Big],$$

(B1)

$$\bar{u}_2 = \frac{Uh - \eta_1\bar{u}_1}{\eta_2},$$

(50)


$$\bar{v}_{1,t} = -\mu\bar{u}_1 - \bar{u}_1\bar{v}_{1,x} + O(\delta^2),$$

(51)

$$\bar{v}_{2,t} = -\mu\bar{u}_2 - \bar{u}_2\bar{v}_{2,x} + O(\delta^2),$$

(52)


$$\zeta_t = (h_1 - \zeta)\bar{u}_{1,x} - \bar{u}_1\zeta_x.$$

(53)

with

$\phi = \frac{1}{\eta_2}\Big[hU_t + U^2 h_x + (\bar{u}_1 - \bar{u}_2)(\eta_1\bar{u}_{1,x} - \bar{u}_1\zeta_x) + \bar{u}_2 U h_x\Big]$

(B2)



$$a(\zeta, h) = 1 + \frac{\delta\eta_2}{1-h}\left[(\eta_2 h_x - \eta_2\zeta_x)(\eta_1/\eta_2)_x - \frac{\eta_2^2}{3}(\eta_1/\eta_2)_{xx} + \frac{\eta_1}{\eta_2}\left(\frac{\eta_2}{2}h_{xx} + \zeta_x h_x\right)\right], \qquad (B3)$$

$$b(\zeta, h) = \delta\left(1 - \frac{\eta_1}{1-h}\right)\eta_1\zeta_x + \frac{\delta\eta_2}{1-h}\left[\frac{\eta_1}{\eta_2}(\eta_2 h_x - \eta_2\zeta_x) - \frac{2\eta_2^2}{3}(\eta_1/\eta_2)_x\right], \qquad (B4)$$

$$c(\zeta, h) = -\delta\left(1 - \frac{\eta_1}{1-h}\right)\frac{\eta_1^2}{3} - \frac{\delta\eta_2}{(1-h)}\frac{\eta_1\eta_2}{3}, \qquad (B5)$$

$$linear = \mu\bar{v}_1 + \zeta_x + \frac{1}{1-h}\left[hU_t + U^2 h_x + \bar{u}_2 h_t - \mu(\eta_1\bar{v}_1 + \eta_2\bar{v}_2) - \eta_1\zeta_x\right], \qquad (B6)$$

$$nonlinear = -\bar{u}_1\bar{u}_{1,x} + \frac{1}{1-h}\left[(\bar{u}_1 - \bar{u}_2)\zeta_t + \bar{u}_1\eta_1\bar{u}_{1,x} + \bar{u}_2\eta_2\bar{u}_{2,x}\right], \qquad (B7)$$

$$dispersive_1 = \delta\left(1 - \frac{\eta_1}{1-h}\right)\left[-\eta_1\zeta_x(\bar{u}_1\bar{u}_{1,xx} - (\bar{u}_{1,x})^2) + \frac{\eta_1^2}{3}(\bar{u}_1\bar{u}_{1,xxx} - \bar{u}_{1,x}\bar{u}_{1,xx})\right], \qquad (B8)$$

$$dispersive_2 = \frac{\delta\eta_2}{1-h}\left[-\eta_2\zeta_x(\bar{u}_2\bar{u}_{2,xx} - (\bar{u}_{2,x})^2) - \frac{\eta_2^2}{3}(\bar{u}_2\bar{u}_{2,xxx} - \bar{u}_{2,x}\bar{u}_{2,xx})\right], \qquad (B9)$$

$$dispersive_{topo} = \frac{\delta\eta_2}{(1-h)}\left[\bar{u}_2 h_x(\eta_2\bar{u}_{2,xx} + \zeta_x\bar{u}_{2,x})\right.$$
$$+ \frac{\eta_2}{2}(U_t h_{xx} + U^2 h_{xxx} + 2U\bar{u}_{2,x}h_{xx} + 2\bar{u}_2 U h_{xxx} + 3\bar{u}_2\bar{u}_{2,x}h_{xx} + \bar{u}_2^2 h_{xxx})$$
$$\left. + \zeta_x(U_t h_x + U^2 h_{xx} + 2\bar{u}_2 U h_{xx} + \bar{u}_2^2 h_{xx})\right]. \qquad (B10)$$



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
