# Peer review of "Limiting amplitudes of fully nonlinear interfacial tides and solitons"

_Nonlinear Processes in Geophysics, 2016_

## Referee Comment (RC1) · Anonymous Referee #1 · 16 Feb 2016

I expected a paper like that a few years earlier, but it happens only now. Sooner or later a semi-analytical baroclinic tidal model for unlimited wave amplitudes should appear. Important is the range of its applicability is wider than just evolutionary stage on free propagating interfacial waves. Unlike the CC theory the presented here model incorporates also the generation stage of internal tides. Ideologically, this approach is similar to Miyata's first theories, but what I can see now is that the model starts with the very beginning of large amplitude internal waves production, when most of the model just fail to work, and I appreciate this fact.

Being a fan of such kind of analytical stuff I just would like to pay some attention to a few specific points that deserve a closer look. Hydrodynamically wise horizontal motions of bottom topography forth and back produce not necessary the same waves as oscillating tidal currents interacting with a motionless sill. Peter Baines did similar experiments and received some critical feedback on this point, but he had no choice trying to reproduce internal tides in laboratory conditions. The authors acknowledge the fact that moving bottom is not the same as a steering tide, line 45-50. They started Section 3 with this statement (lines 291-294) and admit in lines 299-301 that the result could be different in both cases, e.g. tide moving over motionless topography, or generation of internal waves by moving bottom. The difference does really exist. However, making progress we should accept different approaches, so I do not think there is a great difference between two cases, specifically beyond the bottom topography where the "Galilean transformation" (line 299) can be taken into account. However, I really do not understand the reasoning expressed in lines 338-340 about similarity of two coordinate systems with referencing Fig 2. Maybe it is my problem, but I expect some readers can have the same issue. Can the authors justify their point better?

I would also appreciate some sort of revision that would make the paper more oceanographically oriented. Specifically, the parameters of the topography, tidal flow, rotation, etc., - what specific area of the World Ocean the authors have in their mind? Where the effects like that can happen? In terms of the generation mechanism even the Luzon Strait which generates probably the largest internal solitary waves ever recorded shows nearly linear mechanism of internal tide generation over two sills with the Froude number «1. In light of that, I would appreciate any hint on what area of the World Ocean area is targeted? The parameters are described in Figure 2 (see also lines 355-356, Table 1) with h1=30m, h2=70m, and tidal flow 1.2m/sec. Is there any particular object in the World Ocean which is a prototype of that (has I missed something)?

Mathematical procedures are more or less clear, and I trust the authors applied their expansion procedure correctly; I can not raise a red flag at any point. However, there are still a few minor points. The integration through the layers 1 and 2, eqns (19)-(24) looks fine, but I can not say I fully understand Subsection 2.3. In my opinion it is a bit short in explanation of "6 equations and 11 unknowns" although I accept the expansion with respect to delta (depth/wavelength) does can make sense. Some more

details would be necessary to add for better explanation of integral averaging in line 199, as well.

---

## Referee Comment (RC2) · Anonymous Referee #2 · 1 Mar 2016

The work is devoted to a numerical analysis of the MCC-type equations describing strongly nonlinear waves in a two-layer fluid. Its novelties are in adding earth rotation (Coriolis force) and an oscillating forcing imitating a tidal current over a bottom feature. Tidal forcing is represented by an oscillating bottom hill which in most cases gives a reasonable approximation for the case of a fixed hill and periodic current. Ten variants are computed which differ in forcing velocity, layer thicknesses ratio, relative height of the hill, and the Coriolis force (latitude). Some interesting results regarding the parameters of limiting solitons, rate of their formation (in tidal periods). Some results, such as decreasing of soliton amplitude with the increase of forcing, and chande of amplitude and width of a strong limiting soliton, remain unexplained; I agree with authors that it may be due to interaction with current induced by the oscillating source.

In general, the paper deserves publication. However, some questions and notes should

be taken into account. Among them are:

1. The MCC system which is the base of the model, allows strong nonlinearity but only weak (quasi-hydrostatic) dispersion. On the other hand, stationary waves, including a soliton, realize a balance between nonlinearity and dispersion. Thus, (unlike the weakly nonlinear case), applicability of such systems for solitons cannot be taken for granted and need to be verified. It is even possible that some numerical "paradoxes" are due to this limitation (see, e.g., Ostrovsky&Grue, Phys. Fluids, 15, 2934, 2993). This circumstance should at least be mentioned.

2. The weakly nonlinear and "quasi-nonlinear" case is not quite clear for me. It should be close to the eKdV case (where the limiting solitons also exist) but the results seem somewhat different. The physic of this case should be better explained.

3. Paragraph 120. "c0 is an approximate measure of the linear long wave phase speed." – Why approximate, what is the approximation?

4. The reasoning in paragraph 325 should be made simpler and more clear.

5. Figures 8 and 13. How comes that the numerical (color) circles for soliton velocity do not go to 1 at zero amplitude limit? Is this due to some negative period-averaged current?

6. Arguments about the role of higher harmonics are unclear. First, there are no spectra shown in the paper. Second, it remains unclear how the Coriolis dispersion can enhance the table-top soliton form (paragraph 635).

In general: the work is interesting but it is overloaded with details at the expense of clear physical interpretations. If the authors agree to take the above into account, I do not insist on sending the revised paper back to me.

---

## Author Comment (AC1) · 15 Mar 2016

We thank Referee #1 for his/her careful and constructive review on our manuscript. We have uploaded our response as a supplement to this comment. If the editor recommends preparation and submission of a revised manuscript, we will be pleased to incorporate these changes. For clarity in our response, blue font is used for the reviewer's text, black font is used for our text and fonts in *italics* are used for the text that would be included in a revised manuscript.

Please also note the supplement to this comment:
http://www.nonlin-processes-geophys-discuss.net/npg-2016-1/npg-2016-1-AC1-supplement.pdf

**Supplement:**

**Response to Referee 1 (Discussion Forum)**

**Limiting amplitudes of fully nonlinear interfacial tides and solitons (npg-2016-1)**

Borja Aguiar-González[1,2]* and Theo Gerkema[2]

[1]Dpto. de Física, Facultad de Ciencias del Mar, ULPGC, E-35017 Las Palmas, Spain

[2]NIOZ Royal Netherlands Institute for Sea Research, P.O. Box 59, 1790 AB Den Burg, Netherlands

**Anonymous Referee #1**

I expected a paper like that a few years earlier, but it happens only now. Sooner or later a semi-analytical baroclinic tidal model for unlimited wave amplitudes should appear. Important is the range of its applicability is wider than just evolutionary stage on free propagating interfacial waves. Unlike the CC theory the presented here model incorporates also the generation stage of internal tides. Ideologically, this approach is similar to Miyata's first theories, but what I can see now is that the model starts with the very beginning of large amplitude internal waves production, when most of the model just fail to work, and I appreciate this fact.

Being a fan of such kind of analytical stuff I just would like to pay some attention to a few specific points that deserve a closer look. Hydrodynamically wise horizontal motions of bottom topography forth and back produce not necessary the same waves as oscillating tidal currents interacting with a motionless sill. Peter Baines did similar experiments and received some critical feedback on this point, but he had no choice trying to reproduce internal tides in laboratory conditions. The authors acknowledge the fact that moving bottom is not the same as a steering tide, line 45-50. They started Section 3 with this statement (lines 291-294) and admit in lines 299-301 that the result could be different in both cases, e.g. tide moving over motionless topography, or generation of internal waves by moving bottom. The difference does really exist. However, making progress we should accept different approaches, so I do not think there is a great difference between two cases, specifically beyond the bottom topography where the "Galilean transformation" (line 299) can be taken into account. However, I really do not understand the reasoning expressed in lines 338-340 about similarity of two coordinate systems with referencing Fig 2. Maybe it is my problem, but I expect some readers can have the same issue. Can the authors justify their point better?

We agree that our reasoning may be too brief and, therefore, unclear. In a revised version, the following lines would be included in section 3.
* * *
*aguiar@nioz.nl

First, lines 310-312 would read:

'At this point we recall that the oscillation of the topography is included within the forced-MCC equations, in dimensionless form, as $h = h(X)$, with $X(x,t) = x - a\cos t$. The constant $a$ prescribes the strength (speed amplitude) of the oscillating topography via $U = a\,\sin(t)$, the mimicked barotropic tidal flow (see (43)-(47)). For convenience in later discussions, we introduce here $c_T$ to refer the (dimensional) speed of the oscillating topography, which follows from $c_T = c_o\,a$.

Later in the text, lines 338-340 would read:

'In Fig.2, interfacial waves generated from both models are presented for various numerical experiments which differ in the strength of stratification under a fairly strong barotropic tidal flow (Gerkema 1996: gray line) and oscillating topography (forced-MCC: black line). Results over the top of the sill indicate a close correspondence between numerical solutions from Gerkema (1996) and the forced-MCC equations, suggesting only a minor impact of the non-inertial nature of our frame of reference within the parameter space of this study. These results encourage us to refer hereafter the strength (speed) of the oscillating topography as the 'strength of the tidal flow'

I would also appreciate some sort of revision that would make the paper more oceanographically oriented. Specifically, the parameters of the topography, tidal flow, rotation, etc., - what specific area of the World Ocean the authors have in their mind? Where the effects like that can happen? In terms of the generation mechanism even the Luzon Strait which generates probably the largest internal solitary waves ever recorded shows nearly linear mechanism of internal tide generation over two sills with the Froude number 1. In light of that, I would appreciate any hint on what area of the World Ocean area is targeted? The parameters are described in Figure 2 (see also lines 355-356, Table 1) with h1=30m, h2=70m, and tidal flow 1.2m/sec. Is there any particular object in the World Ocean which is a prototype of that (has I missed something)?

We appreciate the interest of the reviewer in knowing whether the present results are applicable to observations in a specific region of the ocean. We have tried to find observational material to compare our findings with, but the difficulty lies in what is actually the strength of the model, namely that it covers all stages, from the creation of the internal waves over topography to the development of the solitons. The problem then is to find observational data on all these stages. We found some on table-top solitons but without the specifics of the source. We would like to continue working on this line and would appreciate it if the reviewer could suggest helpful references. For this paper, we focus on two main goals: first, to present the derivation of a new two-fluid layer model which extends MCC equations with forcing terms and Coriolis effects; and second, to use this novel fully nonlinear model to provide an overview, as generally as possible, on the conditions by which tide-generated interfacial waves may exhibit limiting amplitudes. In line with these two goals, we would add in a revised version the following text in section 5:

*'Whilst not designed to represent a specific region of the ocean, the numerical experiments presented here allow us to investigate for the first time the conditions by which tide-generated interfacial waves may exhibit limiting amplitudes in ocean-like scenarios. With this aim we adopt a two-fluid layer system where the parameters span a broad range of values in order to make clear the qualitative features of these nonlinear processes (Table 1).*

Mathematical procedures are more or less clear, and I trust the authors applied their expansion procedure correctly; I can not raise a red flag at any point. However, there are still a few minor points. The integration through the layers 1 and 2, eqns (19)- (24) looks fine, but I can not say I fully understand Subsection 2.3. In my opinion it is a bit short in explanation of '6 equations and 11 unknowns' although I accept the expansion with respect to delta (depth/wavelength) does can make sense. Some more details would be necessary to add for better explanation of integral averaging in line 199, as well.

We will work on this section and add explanatory comments to clarify the text where necessary.

About the integral averaging in line 199, at the lowest order ($\delta^0$) we are in the hydrostatic regime and horizontal velocities are independent of $z$ within each layer so that $u_i^{(0)} = \bar{u}_i^{(0)}$

$$
\begin{aligned}
\overline{u_i u_i} = \frac{1}{\eta_i} \int dz \, u_i^2 \quad &= \quad \frac{1}{\eta_i} \int dz \, (u_i^{(0)} + \delta u_i^{(1)} + \cdots)^2 \\
&= \quad \frac{1}{\eta_i} \int dz \, (u_i^{(0)\,2} + 2\delta u_i^{(0)} u_i^{(1)} + \cdots) \\
&= \quad u_i^{(0)\,2} + O(\delta) \quad\quad\quad (1) \\
&= \quad \bar{u}_i^2 + O(\delta)
\end{aligned}
$$

so that

$$
\overline{u_i u_i} = \bar{u}_i^2 + O(\delta), \qquad \overline{u_i v_i} = \bar{u}_i \bar{v}_i + O(\delta).
$$

---

## Author Comment (AC2) · 15 Mar 2016

We thank Referee #2 for his/her careful and constructive review on our manuscript. We have uploaded our response as a supplement to this comment. If the editor recommends preparation and submission of a revised manuscript, we will be pleased to incorporate these changes. For clarity in our response, blue font is used for the reviewer's text, black font is used for our text and fonts in *italics* are used for the text that would be included in a revised manuscript.

Please also note the supplement to this comment:
http://www.nonlin-processes-geophys-discuss.net/npg-2016-1/npg-2016-1-AC2-supplement.pdf

[Figure]

**Supplement:**

**Response to Referee 2 (Discussion Forum)**

**Limiting amplitudes of fully nonlinear interfacial tides and solitons (npg-2016-1)**

Borja Aguiar-González[1,2][*] and Theo Gerkema[2]

[1]Dpto. de Física, Facultad de Ciencias del Mar, ULPGC, E-35017 Las Palmas, Spain

[2]NIOZ Royal Netherlands Institute for Sea Research, P.O. Box 59, 1790 AB Den Burg, Netherlands

**Anonymous Referee #2**

The work is devoted to a numerical analysis of the MCC-type equations describing strongly nonlinear waves in a two-layer fluid. Its novelties are in adding earth rotation (Coriolis force) and an oscillating forcing imitating a tidal current over a bottom feature. Tidal forcing is represented by an oscillating bottom hill which in most cases gives a reasonable approximation for the case of a fixed hill and periodic current. Ten variants are computed which differ in forcing velocity, layer thicknesses ratio, relative height of the hill, and the Coriolis force (latitude). Some interesting results regarding the parameters of limiting solitons, rate of their formation (in tidal periods). Some results, such as decreasing of soliton amplitude with the increase of forcing, and chande of amplitude and width of a strong limiting soliton, remain unexplained; I agree with authors that it may be due to interaction with current induced by the oscillating source.

In general, the paper deserves publication. However, some questions and notes should be taken into account. Among them are:

1. The MCC system which is the base of the model, allows strong nonlinearity but only weak (quasi-hydrostatic) dispersion. On the other hand, stationary waves, including a soliton, realize a balance between nonlinearity and dispersion. Thus, (unlike the weakly nonlinear case), applicability of such systems for solitons cannot be taken for granted and need to be verified. It is even possible that some numerical 'paradoxes' are due to this limitation (see, e.g., Ostrovsky & Grue, Phys. Fluids, 15, 2934, 2993). This circumstance should at least be mentioned.

We agree on this important remark, which in a revised manuscript would be discussed in section 5 (Discussion and conclusions), as follows:

'*As previously noted by Ostrovsky and Grue (2003), MCC-type models entail a paradox to the effect that strongly nonlinear solitons appear from a set of equations that have strong nonlinearity but weak dispersion, while the very existence of solitons presume a*
* * *
[*]aguiar@nioz.nl

*balance between the two. In our case, the MCC-type model is used, involving only the lowest-order nonhydrostatic dispersive terms. Despite the small parameter featuring in the nonhydrostatic terms, they may actually become large in practice (i.e., in the numerical runs) if wave profiles are steepening, contradicting the original assumption. Indeed, there is no guarantee that the higher-order dispersive terms, which were dropped from these equations, would always remain small. A suggestion for future work is, therefore, to check our results against a numerical computation with a fully nonlinear nonhydrostatic set of equations.*

2. The weakly nonlinear and 'quasi-nonlinear' case is not quite clear for me. It should be close to the eKdV case (where the limiting solitons also exist) but the results seem somewhat different. The physic of this case should be better explained.

In a revised version, we would make more clear the distinction between the different set of equations being used so that the physical interpretations are also more clear.

The quasi-nonlinear case involves neglecting the baroclinic interactions but retaining the nonlinear terms involving a combination of barotropic and baroclinic fields. The equations are then still linear with regard to the baroclinic fields, but the coefficients become time-dependent due to barotropic factors (which are prescribed), so that higher harmonics will be generated. Hence one should not expect the quasi-nonlinear case to be close to the eKdV on showing limiting interfacial waves.

In a future version we will re-name the quasi-nonlinear interfacial waves as quasi-linear interfacial waves because we understand that the former name has led to confusion when we discussed the physical interpretations and findings.

3. Paragraph 120. 'c0 is an approximate measure of the linear long wave phase speed.' - Why approximate, what is the approximation?

It was not our intention to mean further approximations. The only point is that this quantity is indicative of the phase speed of linear long interfacial waves but not exactly equal to it, for the precise theoretical value has a factor $H_1 H_2/H$, whereas the present factor is $D$. For clarity, we have rephrased this paragraph as follows:

 *'Since we allow waves to have large amplitudes (i.e. being strongly nonlinear), we may take horizontal current velocities to scale with $c_0 = (g'D)^{1/2}$, where $g'$ is reduced gravity, $g' = g \ (\rho_2 - \rho_1)/\bar{\rho}$; and, $c_0$ is close to the linear long-wave phase speed for interfacial waves (which would have $H_1 H_2/H$ instead of $D$). Thus, $u$ and $v$ will be scaled with $c_0$. For the interfacial displacement being allowed to be large, an appropriate scale of $Z$ is $D$.'*

4. The reasoning in paragraph 325 should be made simpler and more clear.

We agree. In a revised version lines 323-330 would read:

*'We use the generation model of weakly nonlinear, weakly nonhydrostatic interfacial waves derived in Gerkema (1996), which works with tidal motion over a fixed topography, as a benchmark for testing the impact of our 'non-inertial' frame of reference. If we compare interfacial waves generated from the nonlinear version of both models, differences are expected to arise from the fact that forced-MCC equations are fully nonlinear. For this reason we restrict the comparison to the linear and quasi-nonlinear cases. If the results between the models turn out to be similar, it thus seems reasonable to conclude that within the framework of study we can compare our present setting to that in the ocean setting.*

5. Figures 8 and 13. How comes that the numerical (color) circles for soliton velocity do not go to 1 at zero amplitude limit? Is this due to some negative period-averaged current?

For clarity, in a revised version Fig. 8 and Fig.13 would also show the corresponding dimensional values along secondary axes (see the new figures at the end of this document).

In all cases the scaled nonlinear phase speed of tide-generated solitons does go to 1 (left $y$-axis) when solitons approach their minimum amplitude, meaning that they are approaching the linear long-wave phase speed for (baroclinic) interfacial waves. Nevertheless, we note that the soliton amplitude of our numerical solutions does never really reach the zero amplitude limit, but it is always above zero. This is because, to get our 'tracking algorithm' working, we need the leading soliton to be large enough so its characteristic points $Z_a$, $Z_b$, $Z_c$ and $Z_d$, as described in Fig. 6, can be identified and used for computation of its amplitude and width. When the solitons are in their very early stage of generation, i. e. amplitudes near zero, the former characteristic points are not well defined yet. As a result, we can't track solitons at the nearly zero amplitude limit.

To account for a more clear and comprehensive discussion of these results, lines 533-543 would read differently in a revised version:

*'Lastly, Fig. 8 compares the wave properties of tide-generated solitons from experiment A1 with solitary wave solutions of the KdV and eKdV theories (Kakutani and Yamasaki (1978), Ostrovsky and Stepanyants (1989), Helfrich and Melville (2006), Gerkema and Zimmerman (2008)).*

*For a fair comparison, we compute the soliton width for KdV and eKdV theories following the same procedure as for the forced-MCC solitons, i. e. we use points $Z_c$ and $Z_d$ (see Fig. 6c-e). The soliton amplitude and width are scaled, respectively, to the thickness of the upper layer and total water depth. The nonlinear phase speed is scaled to the linear long-wave phase speed for (baroclinic) interfacial waves, $c_o' = \sqrt{g' (h_1 h_2)/(h_1 + h_2)}$.*

*In Fig. 8a, small tide-generated solitons approach the linear long-wave phase speed for (baroclinic) interfacial waves, as expected, while larger solitons have a phase speed following a similar curve to that predicted from the eKdV theory. Nevertheless, tide-generated solitons propagate in all cases slower than their eKdV counterparts. Tide-generated solitons ride on interfacial tides and, hence, their wave properties are not simply the response to a settled two-layer fluid system as it occurs for eKdV solitons, but involve also the forcing of the system.*

With respect to Fig. 13, former lines 603-621 would read now simpler:

*'When compared to classical solitary wave theories, tide-generated solitons in experiments B1 and C1 resemble wave properties of strongly and weakly nonlinear solitons, respectively (Fig. 13). As it occurred for experiment A1, tide-generated solitons propagate in all cases slower than their eKdV (experiment B1) and KdV (experiment C1) counterparts of similar amplitude.*

*In experiment B1, the largest solitons start to slightly broaden and present amplitudes larger than those predicted by eKdV theory, although eventually they do not attain the 'table-top' form. In both experiments the largest solitons start to experience a decrease of their amplitude and width when the tidal forcing increases above a certain value, as previously noted from Figs. 11 and 12. These results highlight our finding that tide-generated solitons may be subjected to limiting conditions beyond classical KdV theories. We suggest that these limiting conditions are driven by the appearance of higher harmonics with increasing forcing, and which saturate the underlying quasi-nonlinear internal tide prior to its nonlinear disintegration (see Fig. 3).'*

6. Arguments about the role of higher harmonics are unclear. First, there are no spectra shown in the paper. Second, it remains unclear how the Coriolis dispersion can enhance the table-top soliton form (paragraph 635).

We agree with the reviewer that spectra can be a useful tool to explore higher harmonics, as was done, e.g., in Mercier et al (2012), their Fig. 9. However, we think that our analyses presented in Fig. 3 and Fig. 4 already show convincingly that quasi-nonlinear interfacial tides present limiting amplitudes when the tidal forcing increases, in contrast to the linear regime where higher harmonics are neglected with the increase of the forcing. For clarity, we include in this document a figure where the spectra of power density is shown for several cases. Nevertheless, we prefer not to include these results in the final paper unless this is further recommended. The reason is we think these results may not be sheding more light beyond results from former analyses.

In a revised version, and regarding the mechanism by which the Coriolis dispersion can enhance the 'table-top' soliton form, lines 628-645 would read:

*'Importantly, an unexpected feature must be noted for experiments A1, B1 and C1. At*

*relatively low latitudes off the equator (c. f., numerical solutions for the rotationless case and $\theta{=}15°$), rotational effects appear to favour the development of the leading solitons up to reaching a 'table-top' form. This is especially noticeable in experiments B1 and C1, panels (d) and (f), respectively, for which the rotationless cases do not lead to 'table-top' solitons.*

*In previous sections we have shown that higher harmonics generated by strong tidal flows cause a saturation of the quasi-nonlinear tide by which solitons emerge, and that this factor limits the development of the leading soliton. We suggest that at relatively low latitudes, the saturation of the quasi-nonlinear interfacial tide weakens as the higher harmonics weaken too due to Coriolis dispersion. Accordingly, underlying quasi-nonlinear interfacial tides reach deeper troughs and favour, in the nonlinear regime, leading solitons attaining a 'table-top' form under forcing conditions which would not generate such solitary waves in the rotationless cases. As expected, at higher latitudes ($\theta{=}45°$) the dispersive effect of the Coriolis force becomes stronger, thus not only dispersing the higher harmonics but also preventing the nonlinear interfacial tide from disintegrating into strongly nonlinear solitons.'*

The above phenomenon, occurring at low latitudes, is also illustrated in Fig. 14, where a set of power density spectra (panel b) is shown for the linear, quasi-nonlinear and fully nonlinear interfacial waves of experiment B1 (panel a), without and with rotational effects included ($\theta{=}15°$). Adding rotational effects at low latitudes causes a weakening of the power density of the higher harmonics (c. f., blue and red thick lines in panel b), favouring the rise of quasi-nonlinear interfacial tides with a deeper trough (c. f., blue and red thick lines in panel a). In the nonlinear regime, the addition of rotational effects follows the former pattern (c. f., blue and red thin lines in panels a and b), causing deeper troughs of the underlying quasi-nonlinear interfacial tides which eventually allow the leading solitons to develop a 'table-top' form that does not emerge in the rotationless case. Note here that the 'table-top' form for the rotational case is more evident in Fig. 13c,d, where interfacial waves are 'zoomed in' along the spatial domain.

In general: the work is interesting but it is overloaded with details at the expense of clear physical interpretations. If the authors agree to take the above into account, I do not insist on sending the revised paper back to me.

We agree with Referee #2 on that the discussion of the numerical results could be shortened in order to highlight more the physical interpretations we present. In a revised version we would shorten section 4 following this suggestion.

**References**

*Mercier, M. J., M.Mathur, L.Gostiaux, T.Gerkema, J. M.Magalhaes, J. C. B.DaSilva, and T.Dauxois (2012), Soliton generation by internal tidal beams impinging on a pycnocline: Laboratory experiments, J. Fluid Mech., 704, 37-60.

[Figure]

Figure 8: Solitary wave solutions of the KdV (black line) and eKdV (grey line) theories compared to tide-generated solitary waves derived from the forced-MCC equations (colored dots) for experiments A1. The tide-generated solutions correspond to the mean mature leading soliton propagating in Fig. 7 (experiment A1) along its $4^{th}$ tidal period of age. The different colors indicate the strength of the tidal flow (see legend). (a) Nonlinear phase speed scaled to the linear long-wave phase speed for (baroclinic) interfacial waves vs. soliton amplitude scaled to thickness of the upper layer. (b) Soliton width scaled to total water depth vs. soliton amplitude scaled to thickness of the upper layer. All panels are also shown for the corresponding dimensional form (top and right axes).

[Figure]

Figure 13: Solitary wave solutions of the KdV (black line) and eKdV (grey line) theories compared to tide-generated solitary waves derived from the forced-MCC equations (colored dots) for experiments B1 (top row) and C1 (bottom row). The tide-generated solutions correspond to the mean mature leading soliton propagating in Figs. 11 (experiment B1) and 12 (experiment C1) along its $4^{th}$ tidal period of age. The different colors indicate the strength of the tidal flow (see legend). (a,c) Nonlinear phase speed scaled to linear long-wave phase speed for (baroclinic) interfacial waves vs. soliton amplitude scaled to thickness of the upper layer. (b,d) Soliton width scaled to total water depth vs. soliton amplitude scaled to thickness of the upper layer. The dashed lines in (a) and (c) highlight the varying departure between soliton solutions from eKdV and forced-MCC equations.

[Figure]

Figure 14: (a) Time evolution of linear (L), quasi-nonlinear (QNL) and fully nonlinear (FNL) tide-generated interfacial waves in experiment B1 computed from the forced-MCC-f equations, without and with rotational effects included (blue and red lines, respectively). Spectra of power density for interfacial waves in (a). This time-series corresponds to a 'mooring' located at 115 km leftward from the sill.

---

## Referee Comment (RC3) · Anonymous Referee #3 · 17 Mar 2016

This paper discusses the derivation and then numerical solutions of a fully-nonlinear, weakly-dispersive model for internal tides and solitary-like waves in two-layer stratifications. The model is an extension of the Miyata-Choi-Camassa theory to include rotation and variable topography. While the effects of rotation have previously been studied, the inclusion of variable topography, and forcing of the internal tide by moving topography is new. The authors find that increasing forcing (measured by the maximum speed of the oscillating topography) leads to a maximum amplitude of the radiated internal tide and that further increasing the forcing results in a reduction in radiated amplitude. This is interesting and counter-intuitive result is attributed to the generation of higher harmonics with increasing forcing. Overall the paper contains useful (e.g. the derivation of the model) and interesting results and will be of some interest to the community. However, there are issues with work as presented that need to be addressed. These

are addressed in the comments below.

1. I am not convinced that the model requires the introduction of a moving topography. The authors claim they need to do this to avoid "nonlinearities in the barotropic flow" (line 46). However, they impose a rigid lid and in doing so they can replace $A(t)$ in their equation (48) with $Q(t)$ and set $h_t = 0$. (integrate (46) with $h_t = 0$.) Here $Q(t)$ is a specified, externally imposed barotropic flux. Perhaps this will complicate the equations, but it is possible.

2. In doing what is suggested above, the radiated tides will then be subject to advection by the imposed barotropic flow. This may change the results significantly, especially since they are imposing barotropic flows of order 1m/s in total depths of 100m and the tides and internal waves have speeds of this order. It would certainly call into question the near equivalence of the moving topography and correct barotropic forcing reference frames.

3. The authors discuss a "quasi-nonlinear" version of the model (see line37). However, they never explicitly show the resulting equations, or the precise terms in (41) and (42) that are ignored in this approximation. Further, they never make much of a case as to why one should even explore this aspect. What precisely is learned from this part of the work? How does one connect it to other, mathematically (e.g. asymptotically) consistent models such as the weakly-nonlinear version of (49)-(53) (e.g., the Gerkema and Zimmerman (1995) model). I don't see the value of this part of the analysis.

4. I found the discussion of the numerical experiments very difficult to follow. I was forced to repeatedly go back and forth between Table on and the figures. This was also compounded by the use of dimensional variables. I think that they could simplify the discussion if things are discussed in terms of the governing nondimensional parameters. For example, variations of the reduced gravity $g'$ can be subsumed into a variable relating the timescale of the forcing to the propagation timescale $H/c_0$, where $c_0$ is the linear long wave phase speed. There are of course, other choices, but use of

non-dimensional variables should lead to a more compact discussion and comparison of the cases.

5. The authors claim that the appearance of the saturation in the amplitude of the radiated tide with forcing strength is due to emergence of higher harmonics. While this could be true they never demonstrate it. Furthermore, the emergence of higher harmonic is an indication that the radiated internal tide is itself nonlinear. They might consider that the increased nonlinearity of the radiated tide itself is important. For example, Gerkema and Zimmerman (1995) and Li and Farmer (2011, JPO) discuss the role of weakly-nonlinear internal tide solutions as have Helfrich and Grimshaw (2008) for the fully-nonlinear case considered here. To simply say that higher harmonics is the cause of the maximal response seems to miss the deeper issue. Also, they never show that the same maximal amplitude appears in the full set (49)-(53).

6. Figures 8 and 13 should include the dispersion curves from the Miyata-Choi-Camassa model. After all, this paper is supposed to be about the fully nonlinear waves. Also, some (most?) of the disagreement that is found is likely due to the fact that the solitary waves are propagating on a variable background field (the internal tide). This could be accounted for in the comparison. Note that if the barotropic forcing were included as prescribed time-dependent flux Q(t), then the advection of the solitary waves by the changing barotropic flow would be significant since wave speeds are in the range of 1m/s.

7. The sentence starting on line 625 regarding soliton speeds with rotation is misleading. The soliton speeds are only very weak affected by rotation. However rotation has a large effect on the speed of the internal tide from which the solitons emerge and on which they subsequently propagate ($c^2 = c\_0^2 + f^2/k^2$ in the linear limit).

8. Line 637. The authors never showed that the saturation occurs in the full set of euqations, nor did they demonstrate how it affects the resulting soliton amplitudes.

9. I suggest that the authors remove the linear and quasi-nonlinear results and devote

more effort into exploring the behavior of the fully-nonlinear model. After all, "fully non-linear" is part of the title and the new aspect of the paper. The linear problem has been well covered in the literature and the connection of the "quasi-nonlin ear" reduction with existing weakly nonlinear and now the fully-nonlinear model is not obvious.

---

## Author Comment (AC3) · 11 Apr 2016

We thank Referee #3 for his/her careful and constructive review on our manuscript. We have uploaded our response as a supplement to this comment. If the editor recommends preparation and submission of a revised manuscript, we will be pleased to incorporate these changes. For clarity in our response, blue font is used for the reviewer's text and black font is used for our text.

Please also note the supplement to this comment:
http://www.nonlin-processes-geophys-discuss.net/npg-2016-1/npg-2016-1-AC3-supplement.pdf

**Supplement:**

**Response to REFEREE 3 (Discussion Forum)**

**Limiting amplitudes of fully nonlinear interfacial tides and solitons (npg-2016-1)**

Borja Aguiar-González[1,2*] and Theo Gerkema[3]

[1]Dpto. de Física, Facultad de Ciencias del Mar, ULPGC, E-35017 Las Palmas, Spain.

[2]NIOZ Royal Netherlands Institute for Sea Research, Department of Ocean Systems Sciences

and Utrecht University, P.O. Box 59, 1790 AB Den Burg, Texel, the Netherlands.

[3]NIOZ Royal Netherlands Institute for Sea Research, Department of Estuarine and Delta Systems,

and Utrecht University, P.O. Box 140, 4400 AC Yerseke, the Netherlands.

**Anonymous Referee #3**

This paper discusses the derivation and then numerical solutions of a fully-nonlinear, weakly-dispersive model for internal tides and solitary-like waves in two-layer stratifications. The model is an extension of the Miyata-Choi-Camassa theory to include rotation and variable topography. While the effects of rotation have previously been studied, the inclusion of variable topography, and forcing of the internal tide by moving topography is new. The authors find that increasing forcing (measured by the maximum speed of the oscillating topography) leads to a maximum amplitude of the radiated internal tide and that further increasing the forcing results in a reduction in radiated amplitude. This is interesting and counter-intuitive result is attributed to the generation of higher harmonics with increasing forcing. Overall the paper contains useful (e.g. the derivation of the model) and interesting results and will be of some interest to the community. However, there are issues with work as presented that need to be addressed. These are addressed in the comments below.

1. I am not convinced that the model requires the introduction of a moving topography. The authors claim they need to do this to avoid 'nonlinearities in the barotropic flow' (line 46). However, they impose a rigid lid and in doing so they can replace A(t) in their equation (48) with Q(t) and set $h_t = 0$. (integrate (46) with $h_t = 0$.) Here Q(t) is a specified, externally imposed barotropic flux. Perhaps this will complicate the equations, but it is possible.

As the reviewer already indicates, introducing a barotropic flow in this setting complicates the equations; indeed, the barotropic flow itself would become part of the problem. The point is that one cannot impose a simple barotropic flow in a way that is consistent
* * *
*aguiar@nioz.nl

with the fully nonlinear equations; a barotropic flow would here involve higher harmonics, generated by advective terms like $UU_x$ ($U$ the barotropic flow). In other words, one would actually have to *solve* the barotropic flow from the fully nonlinear equations. Since we are not primarily interested in any intricacies of the barotropic flow, the easier road is here to avoid the problem altogether and prescribe an oscillating topography.

2. In doing what is suggested above, the radiated tides will then be subject to advection by the imposed barotropic flow. This may change the results significantly, especially since they are imposing barotropic flows of order 1m/s in total depths of 100m and the tides and internal waves have speeds of this order. It would certainly call into question the near equivalence of the moving topography and correct barotropic forcing reference frames.

We understand the referee's initial concern in this regard; nevertheless, we consider we have been conservative enough to restrict our study to a parameter space where a semi-equivalence between two different generation models has been tested on the generation of the linear and quasi-linear internal tides (Fig. 2 of the submitted manuscript in the 'Discussion Forum'). If a significant departure between the mimicked tidal flow and the use of an actual tidal flow would exist, it should be then noticeable in the above model-comparison, especially over the top of the oscillating topography; but this was not the case. Far from the sill, the bottom is flat and at rest so it is not expected that the 'non-inertial' frame causes any artifact once the internal tide has been generated and propagates.

Some of the main findings of our study are that quasi-linear tides become saturated as the tidal forcing is increased and that, consequently, leading solitons of the disintegrated internal tides may be also subjected to a limiting amplitude besides that predicted by eKdV and MCC theories. For completeness, and as a double-check, we have tested these findings with the weakly nonlinear model derived in Gerkema (1996), which works with an actual tidal flow over topography. We don't show in this document the full analyses but just a hint of each of them.

Fig. I shows the amplitude saturation of quasi-linear internal tides as the tidal forcing, $c_T$, is increased. As the flow becomes supercritical[1](Fr>1), a further increase of $c_T$ does not generate larger internal tides. This agrees well with our findings from the quasi-linearized version of the forced-MCC equations.

In Fig. II we solve the full set of weakly nonlinear equations derived in Gerkema (1996). Results show how the saturation amplitude of the quasi-linear internal tide, as shown in Fig. I, affects the growth of the leading solitons by also limiting its maximum am-
* * *
[1]To characterize the hydraulic state where internal waves propagate we use the Froude number calculated as $Fr = \frac{c_T}{c_p}$, where the strength of the mimicked tidal flow acting as external forcing, $c_T$, is confronted to the linear long-wave phase speed for interfacial waves, $c_p$.

plitude. The Gerkema (1996) model is built around the weakly nonlinear framework of the classical KdV theory and Klein-Gordon equations, where the amplitude saturation of solitons does not occur. However, Fig. II shows that tide-generated solitons exhibit a limiting amplitude even in the weakly nonlinear regime. Noting this it seems reasonable to argue that the limiting factor is then related to the addition of a tidal forcing.

The above results give support to conclude that findings from the forced-MCC-$f$ equations do not lie on an artifact of the oscillating topography and represent an insightful extension to the fully nonlinear frame of work where tide-generated solitons may attain limiting amplitudes even without reaching a 'table-top' shape, then also subjected to a saturation amplitude of the underlying internal tide prior to its disintegration into solitary waves.

3. The authors discuss a 'quasi-nonlinear' version of the model (see line37). However, they never explicitly show the resulting equations, or the precise terms in (41) and (42) that are ignored in this approximation. Further, they never make much of a case as to why one should even explore this aspect. What precisely is learned from this part of the work? How does one connect it to other, mathematically (e.g. asymptotically) consistent models such as the weakly-nonlinear version of (49)-(53) (e.g., the Gerkema and Zimmerman (1995) model). I don't see the value of this part of the analysis.

In a revised version we will explain more explicitly the distinction between the different set of equations and show in an additional Appendix how the (quasi)-linearization of the forced-MCC-$f$ equations was performed. Also, we will re-name the 'quasi-nonlinear case' as 'quasi-linear case' because we understand that the former name has led to confusion when we discussed the physical interpretations and findings.

For both the forced-MCC-$f$ and Gerkema (1996) models, the quasi-linear case involves neglecting the baroclinic interactions but retaining the nonlinear terms involving a combination of barotropic and baroclinic fields. The equations are then still linear with regard to the baroclinic fields, but the coefficients become time-dependent due to barotropic factors (which are prescribed), so that higher harmonics will be generated. Hence one should not expect the quasi-linear case to be close to the eKdV case on showing saturated interfacial waves.

The above feature is argued in our study to be the most likely factor limiting the growth of leading solitons from the already limited quasi-linear internal tide (besides the soliton saturation predicted by eKdV and MCC theories). This finding is the reason why we find insightful and valuable to start our study on fully nonlinear tide-generated solitons from the generation of the internal tide by which the formers will raise. In a revised version we will make this point more clear as we consider crucial to keep the analyses on the quasi-linear internal tides.

4. I found the discussion of the numerical experiments very difficult to follow. I was

forced to repeatedly go back and forth between Table on and the figures. This was also compounded by the use of dimensional variables. I think that they could simplify the discussion if things are discussed in terms of the governing nondimensional parameters. For example, variations of the reduced gravity $g'$ can be subsumed into a variable relating the timescale of the forcing to the propagation timescale $H/c_0$, where $c_0$ is the linear long wave phase speed. There are of course, other choices, but use of non-dimensional variables should lead to a more compact discussion and comparison of the cases.

In a revised version the discussion of the results will be held using the governing nondimensional variables and the table listing the runs will be presented in a more clear and simplified manner.

5. The authors claim that the appearance of the saturation in the amplitude of the radiated tide with forcing strength is due to emergence of higher harmonics. While this could be true they never demonstrate it. Furthermore, the emergence of higher harmonic is an indication that the radiated internal tide is itself nonlinear. They might consider that the increased nonlinearity of the radiated tide itself is important. For example, Gerkema and Zimmerman (1995) and Li and Farmer (2011, JPO) discuss the role of weakly-nonlinear internal tide solutions as have Helfrich and Grimshaw (2008) for the fully-nonlinear case considered here. To simply say that higher harmonics is the cause of the maximal response seems to miss the deeper issue. Also, they never show that the same maximal amplitude appears in the full set (49)-(53).

We actually think we conclusively demonstrated that the saturation of the amplitude is related to the generation of higher harmonics; this is the very reason why we considered the quasi-linear case in detail. After all, the presence of higher harmonics is the *only* difference between the linear and quasi-linear cases. In the purely linear case, obviously, the solution grows linearly with the forcing. But as the results in Fig. 3 (of the submitted manuscript) show, the quasi-linear case follows the linear growth as long as the barotropic currents are weak, while deviations occur for stronger currents, and then the amplitude becomes saturated. We cannot see any other connection than with the higher harmonics.

6. Figures 8 and 13 should include the dispersion curves from the Miyata-ChoiCamassa model. After all, this paper is supposed to be about the fully nonlinear waves. Also, some (most?) of the disagreement that is found is likely due to the fact that the solitary waves are propagating on a variable background field (the internal tide). This could be accounted for in the comparison. Note that if the barotropic forcing were included as prescribed time-dependent flux Q(t), then the advection of the solitary waves by the changing barotropic flow would be significant since wave speeds are in the range of 1m/s.

We agree. In a revised version the MCC analytical solutions will be included for discussion and comparison with the forced-MCC numerical solutions. Also, we will account for the suggestion made by the referee about the effect of the solitary waves being embedded on a variable background flow, an argument we agree with.

Additionally, in a revised version we will use the Froude number, as define above in item (2), to account for the importance of advection by the changing barotropic flow.

7. The sentence starting on line 625 regarding soliton speeds with rotation is misleading. The soliton speeds are only very weak affected by rotation. However rotation has a large effect on the speed of the internal tide from which the solitons emerge and on which they subsequently propagate ($c^2 = c_0^2 + f^2/k^2$ in the linear limit.

We agree on this important remark that will be corrected in a revised version.

8. Line 637. The authors never showed that the saturation occurs in the full set of equations, nor did they demonstrate how it affects the resulting soliton amplitudes.

Regarding the demonstration of the limiting amplitudes by higher harmonics, we already provided an answer in item (5). About how this amplitude saturation affects the resulting soliton amplitudes, we believe that we have shown numerical solutions doing so. This is described on the basis of presented results, for instance, in lines 578-580 and lines 606-612 of the manuscript submitted to the 'Discussion Forum'. And it is further discussed later in lines 659-670 (Sect. 5. Summary and conclusions) within the scope of main results.

9. I suggest that the authors remove the linear and quasi-nonlinear results and devote more effort into exploring the behavior of the fully-nonlinear model. After all, 'fully nonlinear' is part of the title and the new aspect of the paper. The linear problem has been well covered in the literature and the connection of the 'quasi-nonlinear' reduction with existing weakly nonlinear and now the fully-nonlinear model is not obvious.

Following the referee's request we will devote more effort in a revised version to explore the physical interpretations of the fully nonlinear model by discussing the results using the governing nondimensional parameters (as it was suggested in item (4)).

We understand that in the submitted version to the 'Discussion Forum' it was not clearly explained how the quasi-linearization of the model equations was performed and how that version differs from a weakly nonlinear set of equations. This obviously led to miss an important point of our discussion. This is a topic which we further discussed and answered above in item (3). In a revised version we will make a more clear distinction between the different set of equations.

Regarding the purely linear case, we note this is well covered in the literature. In a revised version we will make more clear that our aim on showing the linear results is only to highlight its departure with the quasi-linear case. It is the latter which presents a new and relevant feature that we investigate, i. e. the saturation amplitude of internal tides subjected to the forcing.

[Figure]

Figure I: Snapshots of the interfacial displacement of leftward propagating quasi-linear internal tides for run A1 ($H_1 = 30$ m; $L_p = 35.49$ km). The amplitude saturation is evident as the tidal forcing is increased and the flow becomes supercritical, $Fr > 1$ (see legend). The run time is 9 tidal periods. The model equations used here are a quasi-linearized version of the weakly nonlinear model in Gerkema (1996).

[Figure]

Figure II: Snapshots of the interfacial displacement of leftward propagating weakly non-linear internal tides and solitons for run A1 ($H_1 = 30$ m; $L_p = 35.49$ km). The limiting amplitude (which is here non 'table-top' shaped) is evident as the tidal forcing is increased but the soliton amplitude becomes saturated. The run time is 9 tidal periods. These waves are generated from the weakly nonlinear generation model derived in Gerkema (1996).

---

## Author Response (AR1)

**Response to the reviews and marked-up version of the revised manuscript**

**Limiting amplitudes of fully nonlinear interfacial tides and solitons (npg-2016-1)**

Borja Aguiar-González[1,2]* and Theo Gerkema[3]

[1]Dpto. de Física, Facultad de Ciencias del Mar, ULPGC, E-35017 Las Palmas, Spain.

[2]NIOZ Royal Netherlands Institute for Sea Research, Department of Ocean Systems Sciences

and Utrecht University, P.O. Box 59, 1790 AB Den Burg, Texel, the Netherlands.

[3]NIOZ Royal Netherlands Institute for Sea Research, Department of Estuarine and Delta Systems,

and Utrecht University, P.O. Box 140, 4400 AC Yerseke, the Netherlands.

**General comment:**

We thank the three referees for their constructive review and thoughtful comments which we have used to improve our manuscript in a revised version. The final author's response to all referee comments (this document) is now provided with reference to all the changes made in the revised manuscript. For clarity, blue font is used for the reviewer's text, black font is used for our response and font in *italics* is used for the new text in the revised manuscript.

As specified in the Journal's instructions, together with the point-by-point response to the reviews we also provide a list of all relevant changes made in the manuscript and a marked-up version of the revised manuscript (all combined in this single *.pdf file). In the marked-up revised manuscript the changes made are in red font where they take only a few lines; when the full section has been rewritten, then the title of the section is in red font. All the changes made refer to comments or clarifications requested by the referees and, therefore, we will refer to those in the point-by-point response to the reviews.

*aguiar@nioz.nl

**List of all relevant changes made in the revised manuscript:**

1. We have re-named the 'quasi-nonlinear case' as 'quasi-linear case' because we understand that the former name led to confusion when we discussed the physical interpretations and findings of our study. Also, we have added for clarity the explicit equations that represent the (quasi-) linearized version of the forced-MCC-$f$ equations in *Appendix B2*.

2. The summary of all runs in Table 1 is now presented in a more compact manner.

3. Figure 5 and related text in the previous manuscript version have been removed in the revised manuscript. Although we think findings from that figure were of interest, we also note they might be out of context for the present study.

4. The validation of the oscillating topography within the parameter space of study is now in *Appendix C*. This helps us to keep fluent the discussion of the main results.

5. Discussion of our results is now based on the governing nondimensional parameters, as suggested by Referee #3. Consequently Sects. 3–6 have been fully rewritten (this is indicated with the title of the section in red font). Our findings keep the same but the discussion of the results is more compact and focused on the most relevant features.

**Response to the reviews**

**Anonymous Referee #1**

I expected a paper like that a few years earlier, but it happens only now. Sooner or later a semi-analytical baroclinic tidal model for unlimited wave amplitudes should appear. Important is the range of its applicability is wider than just evolutionary stage on free propagating interfacial waves. Unlike the CC theory the presented here model incorporates also the generation stage of internal tides. Ideologically, this approach is similar to Miyata's first theories, but what I can see now is that the model starts with the very beginning of large amplitude internal waves production, when most of the model just fail to work, and I appreciate this fact.

Being a fan of such kind of analytical stuff I just would like to pay some attention to a few specific points that deserve a closer look. Hydrodynamically wise horizontal motions of bottom topography forth and back produce not necessary the same waves as oscillating tidal currents interacting with a motionless sill. Peter Baines did similar experiments and received some critical feedback on this point, but he had no choice trying to reproduce internal tides in laboratory conditions. The authors acknowledge the fact that moving bottom is not the same as a steering tide, line 45-50. They started Section 3 with this statement (lines 291-294) and admit in lines 299-301 that the result could be different in both cases, e.g. tide moving over motionless topography, or generation of internal waves by moving bottom. The difference does really exist. However, making progress we should accept different approaches, so I do not think there is a great difference between two cases, specifically beyond the bottom topography where the "Galilean transformation" (line 299) can be taken into account. However, I really do not understand the reasoning expressed in lines 338-340 about similarity of two coordinate systems with referencing Fig 2. Maybe it is my problem, but I expect some readers can have the same issue. Can the authors justify their point better?

We agree that our reasoning in (former) lines 338-340 might be too brief and, therefore, unclear. In the revised manuscript the discussion on the oscillation topography has been reallocated to *Appendix* C, where lines 926–938 make clear our reasoning:

'*In Fig. C.1, interfacial waves generated from both models are presented for various numerical experiments under a fairly strong forcing, i. e. when both models may be expected to deviate more noticeably from each other. Our interest focus then on the upper limit of the supercritical regime (Fr>1) that we can reach while preserving a good agreement between both generation mechanisms. The different settings in Fig. C.1 differ in the strength of stratification from top to bottom panels, while the thickness of the upper and lower layer ($H_1 = 30$ m, $H_2 = 70$ m) and the height and width of the sill are kept fixed ($H_T = 40$ m and $H_L = 10$ km in Eq. 54).*

*Results from Fig. C.1 indicate that in all cases a close correspondence exists between numerical solutions from the model derived in Gerkema (1996) (gray line) and the forced-MCC equations (black line), suggesting only a minor impact of the non-inertial nature of our frame of reference when reaching up to a $Fr\sim1.5$. These results encourage us to approach in our study the strength (velocity amplitude) of the oscillating topography as the 'strength of the tidal flow' within the parameter space of study.'*

I would also appreciate some sort of revision that would make the paper more oceanographically oriented. Specifically, the parameters of the topography, tidal flow, rotation, etc., - what specific area of the World Ocean the authors have in their mind? Where the effects like that can happen? In terms of the generation mechanism even the Luzon Strait which generates probably the largest internal solitary waves ever recorded shows nearly linear mechanism of internal tide generation over two sills with the Froude number 1. In light of that, I would appreciate any hint on what area of the World Ocean area is targeted? The parameters are described in Figure 2 (see also lines 355-356, Table 1) with h1=30m, h2=70m, and tidal flow 1.2m/sec. Is there any particular object in the World Ocean which is a prototype of that (has I missed something)?

We appreciate the interest of the reviewer in knowing whether the present results are applicable to observations in a specific region of the ocean. We have tried to find observational material to compare our findings with, but the difficulty lies in what is actually the strength of the model, namely that it covers all stages, from the creation of the internal waves over topography to the development of the solitons. The problem then is to find observational data on all these stages. We found some on table-top solitons but without the specifics of the source. We would like to continue working on this line and would appreciate it if the reviewer could suggest helpful references. For this paper, we focus on two main goals: first, to present the derivation of a new two-fluid layer model which extends MCC equations with forcing terms and Coriolis effects; and second, to use this novel fully nonlinear model to provide an overview, as generally as possible, on the conditions by which tide-generated interfacial waves may exhibit limiting amplitudes. In line with this, we have added the following text in lines 296–303:

*'Whilst not designed to represent a specific region of the world oceans, we aim to investigate in a general manner the conditions by which tidally-generated solitons may evolve and, eventually, develop limiting amplitudes in ocean-like scenarios. It is then desirable that leading solitons can propagate towards a mature stage before overtaking preceding internal tides; otherwise, although being form-preserving features, the tracking of their wave properties become cumbersome. For this reason the environmental parameters that we describe in the following were selected to highlight the qualitative features of these nonlinear processes for a broad range of (mimicked) tidal forcing strength.'*

And later, in lines 655–657, we have added:

*'Of course the findings presented here cannot describe the whole variety of the specific oceanic conditions. However we believe that this study improves our understanding on the generation and evolution of tide-generated solitons.'*

Mathematical procedures are more or less clear, and I trust the authors applied their expansion procedure correctly; I can not raise a red flag at any point. However, there are still a few minor points. The integration through the layers 1 and 2, eqns (19)- (24) looks fine, but I can not say I fully understand Subsection 2.3. In my opinion it is a bit short in explanation of '6 equations and 11 unknowns' although I accept the expansion with respect to delta (depth/wavelength) does can make sense. Some more details would be necessary to add for better explanation of integral averaging in line 199, as well.

We have rephrased (former) lines 196–198 to clarify the text where we thought it could be of more help. Now lines 199–200 read:

*'Given the z-independence of pressure and returning to the original horizontal momentum equations, it is now natural to assume that the horizontal velocities, too, are independent of z within each layer'*.

About the question of integral averaging in (former) line 199, one must note that at the lowest order $(\delta^0)$ we are in the hydrostatic regime and horizontal velocities are independent of $z$ within each layer so that $u_i^{(0)} = \bar{u}_i^{(0)}$; then,

$$
\begin{aligned}
\overline{u_i u_i} = \frac{1}{\eta_i} \int dz\, u_i^2 \;&=\; \frac{1}{\eta_i} \int dz \left(u_i^{(0)} + \delta u_i^{(1)} + \cdots\right)^2 \\
&=\; \frac{1}{\eta_i} \int dz \left(u_i^{(0)\,2} + 2\delta u_i^{(0)} u_i^{(1)} + \cdots\right) \\
&=\; u_i^{(0)\,2} + O(\delta) \\
&=\; \bar{u}_i^2 + O(\delta)
\end{aligned}
\tag{1}
$$

which yields

$$
\overline{u_i u_i} = \bar{u}_i^2 + O(\delta)\,, \qquad \overline{u_i v_i} = \bar{u}_i \bar{v}_i + O(\delta).
$$

**Anonymous Referee #2**

The work is devoted to a numerical analysis of the MCC-type equations describing strongly nonlinear waves in a two-layer fluid. Its novelties are in adding earth rotation (Coriolis force) and an oscillating forcing imitating a tidal current over a bottom feature. Tidal forcing is represented by an oscillating bottom hill which in most cases gives a reasonable approximation for the case of a fixed hill and periodic current. Ten variants are computed which differ in forcing velocity, layer thicknesses ratio, relative height of the hill, and the Coriolis force (latitude). Some interesting results regarding the parameters of limiting solitons, rate of their formation (in tidal periods). Some results, such as decreasing of soliton amplitude with the increase of forcing, and chande of amplitude and width of a strong limiting soliton, remain unexplained; I agree with authors that it may be due to interaction with current induced by the oscillating source.

In general, the paper deserves publication. However, some questions and notes should be taken into account. Among them are:

1. The MCC system which is the base of the model, allows strong nonlinearity but only weak (quasi-hydrostatic) dispersion. On the other hand, stationary waves, including a soliton, realize a balance between nonlinearity and dispersion. Thus, (unlike the weakly nonlinear case), applicability of such systems for solitons cannot be taken for granted and need to be verified. It is even possible that some numerical 'paradoxes' are due to this limitation (see, e.g., Ostrovsky & Grue, Phys. Fluids, 15, 2934, 2993). This circumstance should at least be mentioned.

We agree on this important remark, which is now discussed in lines 709–719 as follows:

*'Before concluding we must note, as Ostrovsky and Grue (2003) previously did, that fully nonlinear, weakly nonhydrostatic models entail a paradox to the effect that strongly nonlinear solitons appear from a set of equations that have strong nonlinearity but weak dispersion, while the very existence of solitons presume a balance between the two. In our case, the MCC-type model is used, involving only the lowest-order nonhydrostatic dispersive terms. Despite the small parameter featuring in the nonhydrostatic terms, they may actually become large in practice (i.e., in the numerical runs) if internal wave profiles are steepening, hence contradicting the original assumption. Indeed, there is no guarantee that the higher-order dispersive terms, which were dropped from these equations, would always remain small. A suggestion for future work is, therefore, to check our results against a numerical computation with a fully nonlinear nonhydrostatic set of equations.'*

2. The weakly nonlinear and 'quasi-nonlinear' case is not quite clear for me. It should be close to the eKdV case (where the limiting solitons also exist) but the results seem somewhat different. The physic of this case should be better explained.

We have re-named the 'quasi-nonlinear case' as 'quasi-linear case' because we understand that the former name has led to confusion when we discussed the physical interpretations and findings. Also, we have added an explicit explanation with clear distinction on the different set of equations being used. Following the reasoning below, one should not expect the quasi-linear case to be close to the eKdV on showing limiting interfacial waves.

In the revised manuscript lines 357–363 read:

*'The quasi-nonlinear case involves in both generation models neglecting the baroclinic interactions but retaining the nonlinear terms involving a combination of barotropic and baroclinic fields. The equations are then still linear with regard to the baroclinic fields, but the coefficients become time-dependent due to barotropic factors (which are prescribed), so that higher harmonics will be generated when the forcing is increased. For clarification, the (quasi-) linearization of the forced-MCC-f equations is presented in Appendix B2.'*

And later, lines 416–420 read:

*'As described in Sect. 3, we recall that the quasi-linear case includes advective terms from the interactions between the barotropic and baroclinic flows while interactions between baroclinic fields, the genuinely nonlinear terms, are still absent. Therefore, higher harmonics are naturally generated when the forcing is increased. The linear case, where advective terms are absent, is added here for assessing potential departures from the quasi-linear case.'*

3. Paragraph 120. 'c0 is an approximate measure of the linear long wave phase speed.' - Why approximate, what is the approximation?

It was not our intention to mean further approximations. The only point is that this quantity is indicative of the phase speed of linear long interfacial waves but not exactly equal to it, for the precise theoretical value has a factor $H_1 H_2 / H$, whereas the present factor is $D$. For clarity, we have rephrased this paragraph in lines 121-125, as follows:

*'Since we allow waves to have large amplitudes (i.e. being strongly nonlinear), we may take horizontal current velocities to scale with $c_0 = (g'D)^{1/2}$, where $g'$ is reduced gravity, $g' = g\ (\rho_2 - \rho_1)/\bar{\rho}$; and, $c_0$ is close to the linear long-wave phase speed for interfacial waves, $c_p$ (which would have $H_1 H_2 / D$ instead of $D$).'*

4. The reasoning in paragraph 325 should be made simpler and more clear.

We agree. In the revised version lines 918–924 read:

*'We use the generation model of weakly nonlinear, weakly nonhydrostatic interfacial waves derived in Gerkema (1996) (G1996), which works with tidal motion over a fixed topography, as a benchmark for testing the impact of our 'non-inertial' frame of refer-*

*ence. If we compare interfacial waves generated from the nonlinear version of both models, differences are expected to arise from the fact that forced-MCC equations are fully nonlinear. For this reason we restrict the comparison to the linear and quasi-nonlinear model versions. If the results between the models turn out to be similar, it thus seems reasonable to accept that within the parameter space of study we can compare our present setting to that in the ocean setting.*

5. Figures 8 and 13. How comes that the numerical (color) circles for soliton velocity do not go to 1 at zero amplitude limit? Is this due to some negative period-averaged current?

Before addressing this comment, please note that former Figures 8 and 13 are Figure 6 and 11 in the revised manuscript.

In all cases the scaled nonlinear phase speed of tide-generated solitons does go to 1 (left $y$-axis) when solitons approach their minimum amplitude, meaning that they are approaching the linear long-wave phase speed for (baroclinic) interfacial waves. Nevertheless, we note that the soliton amplitude of our numerical solutions does never really reach the zero amplitude limit, but it is always above zero. This is because, to get our 'tracking algorithm' working, we need the leading soliton to be large enough so its characteristic points $Z_a$, $Z_b$, $Z_c$ and $Z_d$, as described in Fig. 6, can be identified and used for computation of its amplitude and width. When the solitons are in their very early stage of generation, i. e. amplitudes near zero, the former characteristic points are not well defined yet. As a result, we can't track solitons at the nearly zero amplitude limit.

6. Arguments about the role of higher harmonics are unclear. First, there are no spectra shown in the paper. Second, it remains unclear how the Coriolis dispersion can enhance the table-top soliton form (paragraph 635).

We agree with the reviewer that spectra can be a useful tool to explore higher harmonics, as was done, e.g., in Mercier et al (2012), their Fig. 9. However, we think that our analyses in Figures 2 and 3 already show convincingly that quasi-linear interfacial tides present limiting amplitudes when the tidal forcing increases, in contrast to the linear regime where higher harmonics are absent. All in all, in the revised manuscript we have decided to take a more conservative position in this regard (see e. g. in lines 9–13). Now we *simply* report that the amplitude limitation of the internal tide occurs with increased tidal forcing when barotropic advection is included to the linear case; and that this appears to be a key factor in the subsequent disintegration of the internal tide into solitons.

In the revised version, and regarding the mechanism by which the Coriolis dispersion can enhance the 'table-top' soliton form, we agree our explanation was unclear. More runs and further analyses are still needed to fully understand the change of shape of the leading solitons and wether they are truly form-preserving as they propagate under these conditions. For this reason we focus now our discussion on the rotational cases on

highlighting the agreement with previous studies, i. e. less solitons at higher latitudes. Thus, lines 631–640 read:

*'In agreement with previous studies we observe in all panels that an increase of the latitude leads to larger dispersive effects due to Coriolis dispersion, which prevents the nonlinear internal tide from disintegrating into strongly nonlinear solitons (Gerkema and Zimmerman 1995, Gerkema 1996). This causes the long internal waves in Fig. 12 to envelope less solitary waves. Also, the internal tides are shown to travel faster as rotation becomes stronger due to rotation increases the phase speed of the linear internal tide, $c_f$ ($c_f^2 = c_0^2 + f^2/k^2$, with k being the wavelength of the internal tide). Although the soliton speeds themselves are only very weakly affected by rotation, they appear traveling faster since they are embedded in the internal tide from which they emerge. As a consequence, leading solitons overtake more quickly preceding internal tides.'*

In general: the work is interesting but it is overloaded with details at the expense of clear physical interpretations. If the authors agree to take the above into account, I do not insist on sending the revised paper back to me.

We agree with Referee #2 on that the discussion of the numerical results could be shortened on details in order to highlight more the physical interpretations we present. In the revised version we have followed this suggestion. Accordingly, Sects. 3–5 have been fully rewritten. Our findings keep the same but the discussion of the results is more compact and focused on the most relevant features.

We understand the referee's initial concern in this regard; nevertheless, we consider we have been conservative enough to restrict our study to a parameter space where a semi-equivalence between two different generation models has been tested on the generation of the linear and quasi-linear internal tides (Fig. C.1 in the revised manuscript). If a significant departure between the mimicked tidal flow and the use of an actual tidal flow

would exist, it should be then noticeable in the above model-comparison, especially over the top of the oscillating topography; but this was not the case. Far from the sill, the bottom is flat and at rest so it is not expected that the 'non-inertial' frame causes any artifact once the internal tide has been generated and propagates.

Some of the main findings of our study are that quasi-linear tides become saturated as the tidal forcing is increased and that, consequently, leading solitons of the disintegrated internal tides may be also subjected to a limiting amplitude besides that predicted by eKdV and MCC theories. For completeness, and as a double-check, we have tested these findings with the weakly nonlinear model derived in Gerkema (1996), which works with an actual tidal flow over topography. We don't show in this document the full analyses but just a hint of each of them.

Fig. I shows the amplitude saturation of quasi-linear internal tides as the tidal forcing, $\mathbf{U_0}$, is increased. As the flow becomes supercritical[1](Fr>1), a further increase of $\mathbf{U_0}$ does not generate larger internal tides. This agrees well with our findings from the quasi-linearized version of the forced-MCC equations.

In Fig. II we solve the full set of weakly nonlinear equations derived in Gerkema (1996). Results show how the saturation amplitude of the quasi-linear internal tide, as shown in Fig. I, affects the growth of the leading solitons by also limiting its maximum amplitude. The Gerkema (1996) model is built around the weakly nonlinear framework of the classical KdV theory and Klein-Gordon equations, where the amplitude saturation of solitons does not occur. However, Fig. II shows that tide-generated solitons exhibit a limiting amplitude even in the weakly nonlinear regime. Noting this it seems reasonable to argue that the limiting factor is then related to the addition of a tidal forcing.

The above results give support to conclude that findings from the forced-MCC-$f$ equations do not lie on an artifact of the oscillating topography and represent an insightful extension to the fully nonlinear frame of work where tide-generated solitons may attain limiting amplitudes even without reaching a 'table-top' shape, then also subjected to a saturation amplitude of the underlying internal tide prior to its disintegration into solitary waves. We include this notion in lines 677–688 of the revised manuscript:

*'Motivated by the above finding we performed analogous runs using the full set of weakly nonlinear equations derived in Gerkema (1996). Because these equations are built around the framework of the classical KdV theory and Klein-Gordon equations, one should not expect that the amplitude saturation of solitons could occur. Nevertheless, results (not shown) demonstrate that both the quasi-linear internal tides and weakly nonlinear tide-generated solitons also exhibit a limiting amplitude. Noting that this model works with an*
* * *
[1]To characterize the hydraulic state where internal waves propagate we use the Froude number calculated as $Fr = \frac{\mathbf{U_0}}{c_p}$, where the strength of the mimicked tidal flow acting as external forcing, $\mathbf{U_0}$, is confronted to the linear long-wave phase speed for interfacial waves, $c_p$. Note that $Fr$ is introduced in the revised manuscript in lines 335–339.

*actual tidal flow over a topography at rest, it seems reasonable to argue that the limiting factor is then related to the addition of a tidal forcing. This gives support to conclude that findings from the forced-MCC-f equations represent an insightful extension to the fully nonlinear frame of work where tide-generated solitons may attain limiting amplitudes with or without reaching a 'table-top' form, then subjected to a saturation amplitude of the underlying internal tide prior to its disintegration into solitary waves.'*

3. The authors discuss a 'quasi-nonlinear' version of the model (see line37). However, they never explicitly show the resulting equations, or the precise terms in (41) and (42) that are ignored in this approximation. Further, they never make much of a case as to why one should even explore this aspect. What precisely is learned from this part of the work? How does one connect it to other, mathematically (e.g. asymptotically) consistent models such as the weakly-nonlinear version of (49)-(53) (e.g., the Gerkema and Zimmerman (1995) model). I don't see the value of this part of the analysis.

In the revised version we have re-named the 'quasi-nonlinear case' as 'quasi-linear case' because we understand that the former name has led to confusion when we discussed the physical interpretations and findings. In Appendix B2 we now show how the (quasi)-linearization of the forced-MCC-$f$ equations was performed.

Additionally, for further clarification, lines 357–363 read:

*'The quasi-nonlinear case involves in both generation models neglecting the baroclinic interactions but retaining the nonlinear terms involving a combination of barotropic and baroclinic fields. The equations are then still linear with regard to the baroclinic fields, but the coefficients become time-dependent due to barotropic factors (which are prescribed), so that higher harmonics will be generated when the forcing is increased. For clarification, the (quasi-) linearization of the forced-MCC-f equations is presented in Appendix B2.'*

And later, lines 416–420 read:

*'As described in Sect. 3, we recall that the quasi-linear case includes advective terms from the interactions between the barotropic and baroclinic flows while interactions between baroclinic fields, the genuinely nonlinear terms, are still absent. Therefore, higher harmonics are naturally generated when the forcing is increased. The linear case, where advective terms are absent, is added here for assessing potential departures from the quasi-linear case.'*

Following the above descriptions one should not expect the quasi-linear case to be close to the eKdV case on showing saturated interfacial waves. We argue in our study this feature is a key factor on limiting the growth of leading solitons besides the saturation predicted by eKdV and MCC theories. This finding is the reason why we find insightful and valuable to start our study on fully nonlinear tide-generated solitons from the generation of the internal tide by which the formers will raise. This motivation is now

made more clear from the very beginning, in the abstract (see lines 9–13), and later in lines 404–420.

4. I found the discussion of the numerical experiments very difficult to follow. I was forced to repeatedly go back and forth between Table on and the figures. This was also compounded by the use of dimensional variables. I think that they could simplify the discussion if things are discussed in terms of the governing nondimensional parameters. For example, variations of the reduced gravity $g'$ can be subsumed into a variable relating the timescale of the forcing to the propagation timescale $H/c_0$, where $c_0$ is the linear long wave phase speed. There are of course, other choices, but use of non-dimensional variables should lead to a more compact discussion and comparison of the cases.

In the revised version the discussion of the results is now held using the governing nondimensional parameters. Sects. 3–6 have been rewritten accordingly. Although the main findings keep the same, we believe the text is now more clear. Also Table 1, listing the runs, is presented in a more simplified manner.

5. The authors claim that the appearance of the saturation in the amplitude of the radiated tide with forcing strength is due to emergence of higher harmonics. While this could be true they never demonstrate it. Furthermore, the emergence of higher harmonic is an indication that the radiated internal tide is itself nonlinear. They might consider that the increased nonlinearity of the radiated tide itself is important. For example, Gerkema and Zimmerman (1995) and Li and Farmer (2011, JPO) discuss the role of weakly-nonlinear internal tide solutions as have Helfrich and Grimshaw (2008) for the fully-nonlinear case considered here. To simply say that higher harmonics is the cause of the maximal response seems to miss the deeper issue. Also, they never show that the same maximal amplitude appears in the full set (49)-(53).

We actually think we conclusively demonstrated that the saturation of the amplitude is related to the generation of higher harmonics; this is the very reason why we considered the quasi-linear case in detail. After all, the presence of higher harmonics is the *only* difference between the linear and quasi-linear cases. In the purely linear case, obviously, the solution grows linearly with the forcing. But as the results in Fig. 3 (of the submitted manuscript) show, the quasi-linear case follows the linear growth as long as the barotropic currents are weak, while deviations occur for stronger currents, and then the amplitude becomes saturated. We cannot see any other connection than with the higher harmonics.

However, in the revised manuscript we have decided to take a more conservative position in this regard (see e. g. in lines 9–13). Now we *simply* report that the amplitude limitation of the internal tide occurs with increased tidal forcing when barotropic advection is included to the linear case; and that this appears to be a key factor in the subsequent disintegration of the internal tide into solitons.

6. Figures 8 and 13 should include the dispersion curves from the Miyata-ChoiCamassa

model. After all, this paper is supposed to be about the fully nonlinear waves. Also, some (most?) of the disagreement that is found is likely due to the fact that the solitary waves are propagating on a variable background field (the internal tide). This could be accounted for in the comparison. Note that if the barotropic forcing were included as prescribed time-dependent flux Q(t), then the advection of the solitary waves by the changing barotropic flow would be significant since wave speeds are in the range of 1m/s.

We agree. In the revised version the MCC analytical solutions are included for discussion and comparison with the forced-MCC numerical solutions (see red curves in Figs. 6 and 11). Also, we have accounted for the suggestion made by the referee about the effect of the solitary waves being embedded on a variable background flow, an argument we agree with (see in lines 557-560).

Additionally, in the revised version we use the Froude number (see Eq. (55)) to account for the importance of advection by the changing barotropic flow.

7. The sentence starting on line 625 regarding soliton speeds with rotation is misleading. The soliton speeds are only very weak affected by rotation. However rotation has a large effect on the speed of the internal tide from which the solitons emerge and on which they subsequently propagate ($c^2 = c_0^2 + f^2/k^2$ in the linear limit.

We agree on this important remark. It is now corrected in lines 631–640:

*'In agreement with previous studies we observe in all panels that an increase of the latitude leads to larger dispersive effects due to Coriolis dispersion, which prevents the nonlinear internal tide from disintegrating into strongly nonlinear solitons (Gerkema Zimmerman 1995, Gerkema 1996). This causes the long internal waves to envelope less solitary waves. Also, the internal tides are shown to travel faster as rotation becomes stronger due to rotation increases the phase speed of the linear internal tide, $c_f$ ($c_f^2 = c_0^2 + f^2/k^2$, with $k$ being the wavelength of the internal tide). Although the soliton speeds themselves are only very weakly affected by rotation, they appear traveling faster since they are embedded in the internal tide from which they emerge. As a consequence, leading solitons overtake more quickly preceding internal tides.'*

8. Line 637. The authors never showed that the saturation occurs in the full set of equations, nor did they demonstrate how it affects the resulting soliton amplitudes.

Regarding the demonstration of the limiting amplitudes, we already provided an answer in item (5). About how this amplitude saturation affects the resulting soliton amplitudes, we believe that we have shown numerical solutions doing so. A summary of the discussion regarding those findings (and numerical solutions) can be found in lines 659–689 of the revised manuscript.

9. I suggest that the authors remove the linear and quasi-nonlinear results and devote more effort into exploring the behavior of the fully-nonlinear model. After all, 'fully nonlinear' is part of the title and the new aspect of the paper. The linear problem has been well covered in the literature and the connection of the 'quasi-nonlinear' reduction with existing weakly nonlinear and now the fully-nonlinear model is not obvious.

Following the referee's request we have devoted more effort in the revised version to explore the physical interpretations of the fully nonlinear model by discussing the results (see Sects. 3–5) using the governing nondimensional parameters; as it was suggested in item (4).

We understand that in the submitted version to the 'Discussion Forum' it was not clearly explained how the quasi-linearization of the model equations was performed and how that version differs from a weakly nonlinear set of equations. This obviously led to miss an important point of our discussion and the reason why the analyses on the quasi-linear case are relevant. This is a topic which we further discussed and answered above in item (3).

[Figure]

Figure I: Snapshots of the interfacial displacement of leftward propagating quasi-linear internal tides for run A1 ($H_1 = 30$ m; $L_p = 35.49$ km). The amplitude saturation is evident as the tidal forcing is increased and the flow becomes supercritical, $Fr>1$ (see legend). The run time is 9 tidal periods. The model equations used here are a quasi-linearized version of the weakly nonlinear model in Gerkema (1996).

[Figure]

Figure II: Snapshots of the interfacial displacement of leftward propagating weakly non-linear internal tides and solitons for run A1 ($H_1 = 30$ m; $L_p = 35.49$ km). The limiting amplitude (which is here non 'table-top' shaped) is evident as the tidal forcing is increased but the soliton amplitude becomes saturated. The run time is 9 tidal periods. These waves are generated from the weakly nonlinear generation model derived in Gerkema (1996).

[revised manuscript text omitted]

---

## Author Response (AR3)

**Response to the Referee #3 and marked-up version of the revised manuscript**
**Limiting amplitudes of fully nonlinear interfacial tides and solitons (npg-2016-1)**

Borja Aguiar-González[1,2]* and Theo Gerkema[3]

[1]Departamento de Física, Facultad de Ciencias del Mar, Universidad de Las Palmas de Gran Canaria, E-35017 Las Palmas de Gran Canaria, Spain.

[2]NIOZ Royal Netherlands Institute for Sea Research, Department of Ocean Systems Sciences

and Utrecht University, P.O. Box 59, 1790 AB Den Burg, Texel, the Netherlands.

[3]NIOZ Royal Netherlands Institute for Sea Research, Department of Estuarine and Delta Systems,

and Utrecht University, P.O. Box 140, 4400 AC Yerseke, the Netherlands.

**General Comment**

The authors' response to Referee #3 is provided with reference to all the changes made in the revised manuscript. For clarity, blue font is used for the reviewer's text, black font is used for our response and font in *italics* is used for the new text in the revised manuscript.

As specified in the Journal's instructions, together with the point-by-point response to the reviews we also provide a list of all relevant changes made in the manuscript and a marked-up version of the revised manuscript (all combined in this single *.pdf file). In the marked-up revised manuscript the changes made are in red font. All the relevant changes made refer to comments or clarifications requested by the referee and, therefore, we will refer to those in the point-by-point response to the reviews.

**List of all relevant changes made in the revised manuscript:**

1. LINES (273–275): '*Far from the sill (i.e., $h \to 0$ for $x \to \pm\infty$), we impose the flow to be purely baroclinic, so that the left-hand side must be zero and hence it follows that $C = 0$.*'

2. LINES (323–325): '*In all experiments, fluid starts moving to the right at $t = 0$ (i.e., topography moving to the left); we start with a system at rest, i.e., $U = \bar{u}_1 = \bar{u}_2 = 0$ at $t = 0$.*

3. LINES (869–871): Because equation $B6$ is called 'linear' while not all terms are in it were linear, we have moved the last three terms from $B6$ in previous versions of the manuscript to $B7$ in the present revised version. Then $B6$ is properly linear. The terms reallocated into $B7$ are in indicated in red font.
* * *
*aguiar@nioz.nl

**Response to the reviews**

**Anonymous Referee #3**

This is a substantially revised and improved version of the manuscript (NPG-2016-1).
The authors have made a thorough attempt to address my previous concerns. However,
I still have two remaining issues that are serious and must be addressed.

The authors' response to my first comment regarding inclusion of the barotropic forcing
is incorrect. Their model employs a rigid lid. As a consequence, $x$-integration of the
continuity equation (45) (in the new manuscript) does give (48) with the undetermined
function of time $C(t)$, as they show. In the case when $h_t = 0$, this function is clearly the
depth-integrated, barotropic, transport. It must be independently specified (because of
the rigid lid). However, it is not necessarily equal to zero for all time simply because
$u_i = 0$ at $t = 0$ as the authors claim (line 273). The authors are making a choice to set
$C(t) = 0$ and oscillate the topography. There is no mathematical or physical problem
with making the opposite choice to take $h_t = 0$ and specify $C(t)$. Both are approxi-
mations to the real ocean. However, the latter is less of an idealization than choosing
the oscillating topography. Specifying the barotropic transport $C(t)$ does not allow for
spatial variations of that transport as would occur with a free surface (in which case an
additional dynamical equation for the barotropic mode would be required). Specification
of $C(t)$ is common in fully-nonlinear models of this type as as, for example, in the work
of Lamb (1994, JGR 99) and the book by Vlasenko et al (2005) (their equation 1.85b
is the equivalent of setting $C(t) = Q_0 \sin(\omega t)$) and related papers. The fact that the
authors have reduced the stratification to two layers and assumed weak non-hydrostatic
dispersion does not change the situation.

Again, I suggest that the authors consider reformulating the problem for an imposed
barotropic flow rather then a moving topography. However, at a minimum they need to
revise their discussion of the issue to take the comments above into account.

We thank the reviewer for his/her comments on this point. The reviewer is right to point
out that our argument following eq. (48) was a *non sequitur*. Contrary to our statement,
the fact that we start with a system at rest has nothing to do with $C(t)$ being zero. We
have corrected this sentence and now say instead:

LINES (273–275): '*Far from the sill (i.e., $h \to 0$ for $x \to \pm\infty$), we impose the flow
to be purely baroclinic, so that the left-hand side must be zero and hence it follows that
$C(t) = 0$.*'

The reviewer suggests that we might take $C(t)$ non-zero, thus imposing a barotropic
tidal flow. This is true, but it would imply a barotropic flow *in addition to a moving*

*topography*, not replacing it, because the very equation (48) is already based on the assumption of a moving topography (in particular, $U$ is its oscillating speed). In other words, if we would want to have a barotropic tide *instead of* an oscillating topography (which would surely be more realistic), we cannot start at eq. (48) but have to return to eqs. (41/42). In that setting, however, the barotropic tide cannot be imposed but *becomes part of the problem* as it has to be solved from the fully nonlinear equations. This is exactly the problem that we have wanted to circumvent by imposing an oscillating topography instead, although this is admittedly less realistic. However, as shown in *Appendix C*, within our parameter settings the deviations stay within acceptable limits.

In the revised version we indicate that we start with a system at rest, now in LINES (323–325): '*In all experiments, fluid starts moving to the right at $t = 0$ (i.e., topography moving to the left); we start with a system at rest, i.e., $U = \bar{u}_1 = \bar{u}_2 = 0$ at $t = 0$.*

The second concern has to do with the quasi-linear model and the interpretation of those calculations. The revised version now makes clear to me what has been done and it then raises new issues. They use this quasi-linear model to argue that the radiated internal tide amplitude reaches a limiting amplitude because of the forcing of the first harmonic. However, from Fig. 2 the departure of the quasi-linear result from the linear calculation occurs when Fr $\approx 1$ and larger. This is just the regime when nonlinearity in the baroclinic mode is $O(1)$. So while there is generation of the first harmonic by the quasi-linear terms (in B11), it is not at all clear that this is a significant factor in the response in the fully-nonlinear model at Fr $\approx 1$. Said another way, I am not convinced that this second harmonic from the quasi-nonlinear terms is more important than other nonlinear effects (that will also result in higher harmonics) that occur when Fr $= O(1)$. Lastly, while it may be a fine point considering the preceding discussion, I do not see a clear limiting amplitude in the quasi-linear results in Fig. 5. Neither a local maximum nor a well-define asymptote is obvious. If it does occur is must be for Fr   1.6, even further into the regime where the quasi-linear model and interpretation is suspect.

I also suggest strongly that they remove the material on the quasi-linear calculations. The results are suspect and potentially misleading regarding the source of any nonlinearity in radiated baroclinic mode when Fr $= O(1)$.

We thank the reviewer for his/her comments on this point. However, according to the following arguments we consider it important to retain the results and discussion on the quasi-linear calculations.

It's true that to show the maximum amplitude in Fig. 2 much more clearly for all experiments we might need to go for larger Fr in some of them (in $C1$ and $C2$ for instance) but we think that it is clear enough in all the other experiments, thus, supporting validity of the discussed feature, i.e. quasi-linear internal tides do not simply grow linearly but approach a limiting amplitude as the forcing increases. As the referee notes, we must restrict ourselves to a 'safe' parameter space. However, we do not claim at any

point that the limiting amplitude of quasi-linear internal tides follows an asymptotical behavior as the forcing increases (we cannot reach Fr larger than 1.6 with our model and, consequently, we do not discuss that parameter space).

We must also note that in the quasi-linear calculations not only the first and second harmonics are present, as the referee mentions, but also higher harmonics emerge. We agree with the reviewer that when the fully nonlinear model is acting, we cannot quantify/separate the relative effect of higher harmonics generated by the forcing in the quasi-linear regime and higher harmonics generated by other nonlinear terms in the fully nonlinear regime. This is the reason why we need to split the analysis in two stages, i. e., to better understand the underlying processes. This is in our view a natural way to study the process, since the generation of the underlying internal tide can be understood from quasi-linear theory, from which solitons emerge only later as nonlinear effects become increasingly important. This procedure of first studying the parameter dependence in the generation of the internal tide and, subsequently, its disintegration is common in the literature, see,. e.g. Sandstrom & Quon (1993), Sandstrom & Quon (1994) and Gerkema (2001), as we already explained in the manuscript (LINES 409–414). We think we have shown the outcome of this analyses is revealing on what happens in the fully-nonlinear model.

However, in response to the reviewer's comments, we do not attribute any longer the limiting amplitude of the internal tide and solitons to the appearance of higher harmonics. Now we only discuss that a limiting amplitude of the quasi-linear tide appears under increasing barotropic forcing and that this preconditioning *may* be responsible of generating saturated solitons before reaching their theoretical maximum amplitude (see LINES 7–13).

**References**

[revised manuscript text omitted]

---

## Author Response (AR4)

**Response to the Referee #3 and marked-up version of the revised manuscript**

**Limiting amplitudes of fully nonlinear interfacial tides and solitons (npg-2016-1)**

Borja Aguiar-González[1,2]* and Theo Gerkema[3]

[1]Departamento de Física, Facultad de Ciencias del Mar, Universidad de Las Palmas de Gran Canaria, E-35017 Las Palmas de Gran Canaria, Spain.

[2]NIOZ Royal Netherlands Institute for Sea Research, Department of Ocean Systems Sciences

and Utrecht University, P.O. Box 59, 1790 AB Den Burg, Texel, the Netherlands.

[3]NIOZ Royal Netherlands Institute for Sea Research, Department of Estuarine and Delta Systems,

and Utrecht University, P.O. Box 140, 4400 AC Yerseke, the Netherlands.

**General Comment**

The authors' response to Referee #3 is provided with reference to all the changes made in the revised manuscript. For clarity, blue font is used for the reviewer's text, black font is used for our response and font in *italics* is used for the new text in the revised manuscript.

As specified in the Journal's instructions, together with the point-by-point response to the reviews we also provide a list of all relevant changes made in the manuscript and a marked-up version of the revised manuscript (all combined in this single *.pdf file). In the marked-up revised manuscript the changes made are in red font. All the relevant changes made refer to comments or clarifications requested by the referee and, therefore, we will refer to those in the point-by-point response to the reviews.

**List of all relevant changes made in the revised manuscript:**

1. Relevant changes refer to the two main issues discussed by the referee and are addressed in our response with clear indication (reference to LINES) of where these changes have been made in the text.

*aguiar@nioz.nl

**Response to the reviews**

**Anonymous Referee #3**

*The authors are wrong about the barotropic flow issue. If the assumption of a rigid lid is removed, then the barotropic flow must be solved for as part of the problem. But with a rigid lid the barotropic flow must be specified. Their equation 48 with the topographic motion set to zero (U=0, as my most recent review suggested that they do, but still retaining a stationary topography) is still correct and C(t) is the barotropic mass flux. For the moment consider the shallow water limit (the nonhydrostatic dispersion terms are zero) then their equations (41), (42) and (48) (with U=0) are equivalent to (2.1) − (2.3) in Sandstrom and Quon (1993), where the barotropic flux is imposed (q(t) in that paper). The addition of the nonhydrostatic terms does not change this (see my earlier review comment on barotropic flux in Lamb (1994, JGR 99) and the book by Vlasenko et al (2005)). The authors seem to implicitly understand this as they impose a purely baroclinic flow in the far field (lines 273–275). They are specifying the barotropic flux! Setting it to zero is simply a specific choice of C(t), but not the only one. As before, I think the authors just need to discuss briefly that they choose to oscillate the topography instead of imposing the barotropic flux through C(t) and that with a rigid lid there is no need to solve independently for the barotropic flow.*

We thank the reviewer for his/her comments. We now see his/her point more clearly. As suggested by the reviewer, we have briefly discussed in the manuscript our choice of a moving topography and the alternative approach proposed by the referee.

LINES 46–54: '*To avoid the need to solve nonlinearities from the barotropic tide itself (which cannot be formally neglected in a fully nonlinear model), we mimick the interfacial wave generation by barotropic tidal flow over topography with a horizontally oscillating topography. (There is no complete equivalence with an oscillating flow, but we demonstrate that for the parameters used here, the difference remains small.)*

*The presence of a topography greatly complicates the subsequent handling of the equation, necessary to bring them in a form amenable to numerical solving, but we demonstrate that the set of equations can be obtained. An alternative approach will be also discussed later.*'

LINES 298–308: '*Before concluding this section, it is worth while noting an alternative approach. Given the assumption of a rigid lid, one could have also taken $U = 0$ in (48), the topographic motion set to zero, and then prescribe an external barotropic flux via $C(t)$. Imposing a barotropic flux in this manner does not allow for spatial variations of that flux as it would occur with a free surface, for which an additional dynamical equation would be required to solve the barotropic mode. Specification of $C(t)$ is common in fully*

*nonlinear models of this type as, for example, in Lamb (1994) and Vlasenko et al (2005). However, the choice of an oscillating topography has also proven to be of use on the study of strongly nonlinear interfacial waves. For instance, Grue (2015) recently confirmed findings on the onset of wave train formation observed in experimental measurements by Maxworthy (1979) with a three-dimensional two-layer, fully dispersive and strongly nonlinear interfacial wave model with a time-varying bottom topography.'*

Regarding the second point: They say in the abstract 'we use the model equations to investigate the role of the initial stages of the internal tide on limiting the amplitudes of solitons under fully nonlinear conditions.' My point was that for Fr = O(1) the initial stages (i.e. the internal tide generated by the topography) is a fullynonlinear problem. The Sandstrom and Quon papers do break the problem up, and I agree that a unified approach is preferred. However, Sandstrom and Quon use the shallow water model (fully nonlinear) to study the generation, and then the weakly nonhydrostatic model for the far field. The authors should at least show the radiated internal tide from the fully nonlinear and the quasilinear models agree with each other in the immediate neighborhood of the topography for the Fr = 0.5–1.6 range. If they do not agree, then the quasilinear results are not directly relevant.

We thank the reviewer for his/her comments. We understand that maybe citing the Sandstrom and Quon papers was misleading in this regard as the comparison is not straightforward. Accordingly, we have decided to remove that piece of text and now LINES 419–422 read:

'*Tide-generated solitons emerge from nonlinear disintegration of the underlying internal tides and may be, therefore, naturally subjected to their wave properties. For this reason, we find it insightful to investigate first the wave properties of the underlying internal tides, prior to its nonlinear disintegration, within the parameter space of this study.*'

Regarding the request of testing whether the fully nonlinear and the quasilinear models agree with each other in the immediate neighborhood of the topography for the range of study, we present the corresponding analysis in Fig. I. We think the results are quite convincing on showing a good agreement between the troughs shaping the recently generated quasi-linear and fully nonlinear interfacial waves. It is along these troughs that nonlinearities quickly arise in the fully nonlinear model as one would expect. We use the quasi-linear analyses and findings in our work as suggestive, or potentially explanatory, of what happens in the fully nonlinear case.

Also the abstract now reads slightly different to account for the comment of the reviewer and make the reader aware about how we proceed. Now LINES 7–13 read: '*Besides, we use the quasi-linearized model equations to investigate the role of the initial stages of the internal tide prior to its nonlinear disintegration. Numerical solutions reveal that the internal tide, considered linear but with the inclusion of barotropic advection (the quasi-linear case), reaches a limiting amplitude under increasing barotropic forcing.*

*Numerical experiments in the fully nonlinear regime suggest that this limiting amplitude in the underlying internal tide extends to the nonlinear case in that internal solitons formed by a disitintegration of the internal tide may not reach their table-shaped form with increased forcing but appear limited well below that state.'*

[Figure]

Figure I: Quasi-linear (black line) and fully nonlinear (grey line) interfacial waves generated over an oscillating sill from the forced-MCC-$f$ equations (run A1). The Froude number and corresponding strength of the (mimicked) tidal flow are indicated in the upper-right corner of each panel. For scaling purposes one must note that the wavelength of the linear long-wave interfacial wave is $L_p = 35.49$ km ($g' = 0.03$ m s$-1$) while the thickness of the upper layer is $H_1 = 30$ m. The run time is 9 tidal periods.

**References**

[revised manuscript text omitted]